



# Storm surge hazard over Bengal delta: A probabilistic-deterministic modelling approach

Md Jamal Uddin Khan[1,3], Fabien Durand[1,2], Kerry Emanuel[4], Yann Krien[5], Laurent Testut[1,3], and A.K.M. Saiful Islam[6]

[1]LEGOS UMR5566, CNRS/CNES/IRD/UPS, 31400 Toulouse, France
[2]Laboratório de Geoquímica, Instituto de Geociencias, Universidade de Brasilia, Brazil
[3]UMR 7266 LIENSs, CNRS- La Rochelle University, 17000 La Rochelle, France
[4]MIT, USA
[5]SHOM, DOPS/STM/REC, Toulouse, France
[6]IWFM, BUET, Dhaka 1000, Bangladesh

**Correspondence:** Md Jamal Uddin Khan (jamal.khan@univ-lr.fr)

**Abstract.** Storm-surge induced coastal inundation constitutes a substantial threat to lives and properties along the vast coastline of the Bengal delta. Some of the deadliest cyclones in history made landfall in the Bengal delta region claiming more than half a million lives over the last five decades. Complex hydrodynamics and observational constraints have hindered the understanding of the risk of storm surge flooding of this low-lying (less than 5m above mean sea level), densely populated (>150M) mega-

delta. Here, we generated and analysed a storm surge database derived from a large ensemble of 3600 statistically and physically consistent synthetic storm events and a high-resolution storm surge modelling system. The storm surge modelling system is developed based on a custom high-accuracy regional bathymetry enabling us to estimate the surges with high-confidence. From the storm surge dataset, we performed a robust probabilistic estimate of the storm surge extremes. Our ensemble estimate shows that there is a diverse range of water level extremes along the coast and the estuaries of the Bengal delta, with well-defined

regional patterns. We confirm that the risk of inland storm surge flooding at a given return period is firmly controlled by the presence of coastal embankments and their height. We also conclude that about 10% of the coastal population is living under the exposure of a 50-year return period inundation under current climate scenarios. In the face of ongoing climate change, which is likely to worsen the future storm surge hazard, we expect our flood maps to provide relevant information for coastal infrastructure engineering, risk zoning, resource allocation, as well as future research planning.

## 1   Introduction

Among global coastlines exposed to storm surges, the countries along the Indian Ocean have some of the most severely impacted places by deadly cyclones (Needham et al., 2015). Particularly the northern Bay of Bengal is one of the deadliest regions in terms of cyclone related mortality (Ali, 1999). Although a tiny percentage (~6%) of cyclones forms and makes landfall around this region, the total death toll is more than 50% of the global total (Alam and Dominey-Howes, 2014). The

Bay of Bengal experiences, on average, five surge events per decade exceeding a surge level (excess water level above the tidal prediction) of 5m (Needham et al., 2015). The Bengal delta coastline, located in the Northern Bay of Bengal spanning

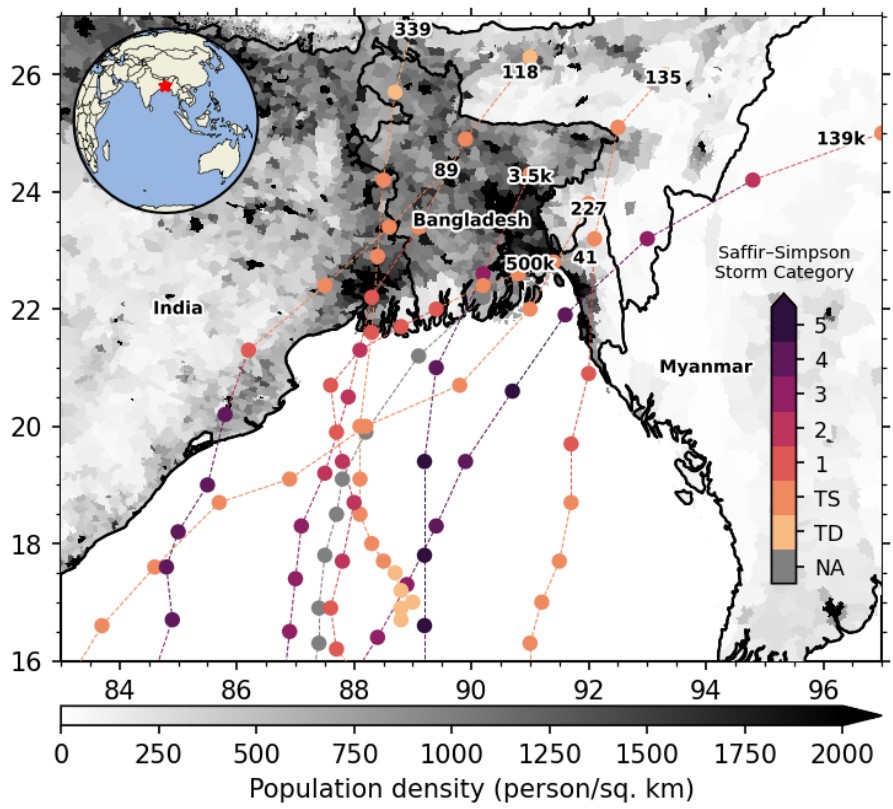

**Figure 1.** Population density in the Bengal delta covering Bangladesh and India. A subset of cyclones tracks that made landfall in past decades is displayed in dashed lines, with the number indicating the associated casualty and the colour indicating the strength of the cyclone. Inset shows the location of the study area. TD and TS correspond to Tropical Depression and Tropical Storm, respectively. The colour-coded lines show the cyclone strength along its track in the Saffir-Simpson scale. For the 1970 Bhola cyclone, the wind speed is not available and shown in grey (NA). The associated number at the head of the cyclone tracks refers to the number of casualties.

Bangladesh and India, experiences a large portion of the aforementioned cyclonic storms and associated storm surges. More than 100 million people live below 10m elevations in this densely populated mega delta region (Becker et al., 2019). Unsurprisingly, in the Bengal delta region, higher surge levels have a statistically significant correlation with the number of casualties in the past cyclones (Seo and Bakkensen, 2017).

Figure 1 illustrates a representative subset of the deadly cyclones that occurred in the last five decades. Table 1 lists the illustrated subset of cyclones and associated surges. This list is not exhaustive and is only shown here to illustrate the storm surge hazard. For a comprehensive historical catalogue of cyclones over the Bengal delta region see Alam and Dominey-Howes (2014).

This small sample of cyclones and their associated surge indicates that cyclones can induce strong storm surge along this macro-tidal region of the world, depending on the tidal condition on their landfall. The regional characteristics of the tide are





predominantly semi-diurnal in this region (Sindhu and Unnikrishnan, 2013). The tidal range is high, reaching about 5m along the north-east coast (Krien et al., 2016; Khan et al., 2020). Additionally, a sizable continental river flow discharges through the distributaries of Ganges-Brahmaputra-Meghna (Mohammed et al., 2018). The shallow coastal submarine delta with an average

width of 50km provides favourable conditions for the development of large surges and strong interaction with tide (Krien et al., 2017b). The combination of all the aforementioned properties can give rise to high storm-driven water levels that have the potential to inundate the low-lying delta region and to cause thousands of casualties (Murty et al., 1986). Considering the tidal range and strong tide-surge interaction, the term "water level" is hereafter used to refer to the water level derived from the dynamic interaction of tide, surges and waves.

Although we have opened our problem statement highlighting the deadliness of the cyclones landfalling in the Bengal delta region, the number of casualties from the storm surge has been reduced drastically over the last decades (Paul, 2009). This improvement is partly due to the governmental effort to build cyclone shelters and pre-storm preparedness. The local government and communities put substantial efforts to implement timely evacuations during storms. Continued development in the communications system (e.g. mobile phone network) and improvement in the numerical storm and surge forecasting further

facilitated the evacuation effort. Engineering structures like embankments, which were built to promote agriculture through controlling tidal flooding (Nowreen et al., 2013), also provided a significant level of protection during storm surge events (Adnan et al., 2019). For a recent storm, supercyclone Amphan, Khan et al. (2021a) shows that the major mode of flooding was due to failure of the dikes, indicating either inadequate design, or poor maintenance, or both. Confident engineering design and continued maintenance of these structures, however, depends on the understanding of the variability of sea level, particularly

the extreme water levels (Chowdhury et al., 1998).

Unfortunately, the variability of water level along the intricate coastline of the Bengal delta is poorly observed (Woodworth et al., 2016). The available limited set of water level observations reveals regionally varying water level dynamics at long, as well as short timescales (Tazkia et al., 2017; Becker et al., 2020; Antony et al., 2016). For instance, Antony et al. (2016) found a robust amplification of extreme high-water levels from south to north along the east coast of India. In their analysis,

the tide appears as the largest contributor to extreme high-water events. At seasonal timescale, Tazkia et al. (2017) showed a summer-winter variability of mean sea level along the Bengal delta coastlines reaching up to 70cm in range. At decadal scale, Becker et al. (2020) showed a consistent increase in relative sea level over the delta from an extensive database of daily water level records.

Nevertheless, due to the limited availability of water level records, consistent estimation of water level hazard for storm

surge from in situ observations is not yet realized in this region (Lee, 2013; Chiu and Small, 2016). The main challenge is to secure long-enough hourly (or higher temporal resolution) tide gauge records for reliable statistical analysis (Unnikrishnan and Sundar, 2004). The separation of short timescale (driven by meteorological effects, among which surges) from longer timescale (driven by tidal effects) on water level variability poses another challenge. The failure of tide gauges during a cyclonic event is also commonplace, causing bias in the record of extreme water levels (Antony et al., 2016; Krien et al., 2017b; Chiu and

Small, 2016).



Hydrodynamic modelling of the storm surges has been used, with varying degree of complexity, to curb the aforementioned data unavailability problems - along both the east coast of India (Chittibabu et al., 2004; Jain et al., 2010; Sindhu and Unnikrishnan, 2011) and the coast of Bangladesh (Chowdhury et al., 1998; Jakobsen et al., 2006).

Along the east coast of India, the approach taken by Chittibabu et al. (2004) and that of Jain et al. (2010) are similar. Their
estimation of the 50-year return period water level involved determining a 50-year return level of the atmospheric pressure drop ($\Delta P$) from the observed cyclone records. Afterwards, they used the pressure drop estimate in conjunction with artificial cyclone tracks to spatially cover the area of interest. In their approach, the tide is added linearly to the cyclone surge. For example, Chittibabu et al. (2004) estimated the 50-year return period tidal water level and added it with the 50-year surge water level. On the other hand, Jain et al. (2010) added the total tidal amplitude for M2, S2, K1 from FES2004 global tidal atlas
(Lyard et al., 2006) to the estimated surge water level. Both studies considered an additional constant water level of 50cm to account for the wave set-up. This approach by Chittibabu et al. (2004) and Jain et al. (2010) assumes that tide and surge do not have any interdependence and that 50-year return level tide and 50-year return level surge adds up to 50-year return level tide-surge. Moreover, the wave set-up is assumed to be the same for all return periods, in the sense that the linear addition of the wave-setup does not change the return period of an estimated water level. For the Orissa coast in India (19°N-21°N), they
both obtained a similar pattern of combined total water level (combining surge, tide, and wave set-up) at a 50-year return period with an increase from 4.3m in the south to 8.5m in the north. The numbers reported in Jain et al. (2010) (see their Figure 10) seem to be the same as those of Chittibabu et al. (2004) (see their Figure 8b). Along the coastline of West Bengal (21°N-20°N), the estimated 50-year return level ranges from 5 to 10m.

Sindhu and Unnikrishnan (2011) addressed the non-linear dependence of tide and surge with a coupled tide-surge model.
In their analysis, Sindhu and Unnikrishnan (2011) simulated 136 events spanning 1974 to 2000 to estimate maximum water levels during those storms. They applied an extreme value analysis on the estimated water levels using an r-largest annual maxima approach to determine the return period of extreme water levels. Similar to Chittibabu et al. (2004) and Jain et al. (2010), Sindhu and Unnikrishnan (2011) also found an increasing amplitude of 50-year return level of total sea level from south to north along the east coast of India. The highest 50-year return period water level was estimated in the northern edge
– Sagar Island (88.11°E, 21.72°N) (8.7 m) and Chandipur (87.01°E, 21.43°N) (6.9m). Compared to the 50-year return water level estimates of Chittibabu et al. (2004) and Jain et al. (2010), the estimates by Sindhu and Unnikrishnan (2011) are inferior in water level by about 50% throughout the coast of Orissa. One notable issue with the approach of Sindhu and Unnikrishnan (2011) is, however, the limited number of available storm events. They derived the extreme water level along the 1200km shoreline from 136 storm events, i.e., three events per 30 kilometres (typical footprint of storm surges). This spatial sampling is
potentially not adequate to capture the storm surge water level at various phases of the tide. Additionally, the storm parameters are collected over a long period, and the homogeneity of the storm records is not well defined (Singh et al., 2020). Finally, these studies focused on the east coast of India do not cover the Bangladesh part of the delta, where a larger tide and a stronger tide-surge interaction is observed (Antony and Unnikrishnan, 2013).

In the Bangladesh part of the delta, Chowdhury et al. (1998) estimated the water levels inside the estuaries with a 1-D
model at a 10- to 50-year return period to plan the construction of cyclone shelters. In their approach, they used an empirical





formulation to estimate the surge level for a given wind speed. To estimate the surge level at a given return period, they assumed that the surge height corresponds to the wind speed of the approaching cyclone to that location at half the return period. This relatively simple approximate solution holds on this macro-tidal regime. Statistically, the tidal water level is relatively high for about half of the cyclones - those making landfall during the high-tide part of the tidal cycle. Their estimated 50-year return period surge level ranges from 3 to 4m along the central part of the delta from west to east. However, in the final estimate for the design water level for the cyclone shelters in their study, they added the spring tide level linearly. They did not, however, provide an estimate of the total water level at the 50-year return period.

Jakobsen et al. (2006) took an approach similar to Sindhu and Unnikrishnan (2011) to estimate the extreme water level. For the Bengal delta coastline (of about 300km length) they selected 17 historic cyclones that produced significant surges during 1960-2000. The simulated 17 water level records at each model grid point were then fitted to an exponential distribution to estimate the associated distribution parameters. According to their estimate, the 100-year return period water level goes as high as 10m at the northeastern corner of the estuary. They also noted a rapid evolution of water level with return periods along the shoreline of the lower part of the Meghna estuary.

From the literature review, it is clear that the unavailability of data also hindered the previous attempts to map the storm surge hazard using numerical models. This lack of data includes both the storm track records and associated storm parameters – e.g., pressure drop ($\Delta P$), maximum wind speed ($V_{max}$), and radius of maximum wind ($R_m$). The recorded storm parameters also have inherent inhomogeneity due to gradual changes in observational techniques (Singh et al., 2020). Finally, the spatial and temporal coverage of the recorded storms is sparse relative to the size of the shoreline and sampling of tidal phases. Indeed, the past studies reported sensitivity of the surge level estimates to storm parameters around the Bengal delta region, both for the maximum water level (~O(m)) and the inundation extent (~O(km)) (Lewis et al., 2014; Hussain et al., 2017).

Along with data problem, the past modelling systems suffered systematic error due to modelling simplification. Notably, the past studies often did not consider the tide-surge interactions or wave set-up dynamically, e.g., Chittibabu et al. (2004) and Jain et al. (2010). Over the last decade, hydrodynamic modelling in this region saw significant progress. Thanks to the improvement of bathymetry and topography, tidal circulation is now comparatively well reproduced (Krien et al., 2016; Khan et al., 2019). On top of it, the implementation of coupled hydrodynamic-wave model for cyclones has improved the realism of the extreme water level dynamics during storms (Krien et al., 2017b; Murty et al., 2016). These improvements in modelling naturally call for revisiting the prospect of understanding the risk of flooding from cyclone induced storm surges. The last important piece in this risk assessment is the climatology of the storms.

Our understanding of the dynamics of the cyclones has also progressed over the last decade. Recent advancement in this field enabled translating climatic properties to large ensembles of statistically consistent storm events. Examples of such methods include distribution sampling (Rumpf et al., 2007), Joint Probability Method with Optimal Sampling (JPM-OS) (Toro et al., 2010) and statistical-deterministic approach (Emanuel et al., 2006). Among these approaches, the statistical-deterministic approach is perhaps the most advanced. This modelling approach combines the statistical generation of the storms with a deterministic evolution and intensity estimation. The result is an ensemble of physically consistent cyclones representing the cyclonic activ-





ity of a basin climate. The statistical-deterministic approach has been successfully used to estimate the storm surge hazard at a local scale (Lin et al., 2010; Krien et al., 2017a, 2015), as well as at continental scale (Haigh et al., 2013).

It is noteworthy here that in a recent work by Leijnse et al. (2021, under review, last accessed 27 September 2021), a combination of statistical storm modelling and hydrodynamic simulation is employed to study the coastal hazard over Bay of Bengal. They modelled 1000 years of storm activity with TCWiSE (Tropical Cyclone Wind Statistical Estimation), and

simulated them with a coarse resolution unstructured grid hydrodynamic model (highest resolution at 3km). Their results indicate a substantial reduction in the uncertainty in the hazard estimate from the synthetic cyclones compared to the observed storm dataset. However, the major limitation of their model is the use of the globally available bathymetry in the nearshore zone, which is found to have substantial bias and consequently a significant impact on the tidal hydrodynamics (Krien et al., 2016). The authors also acknowledge that the kilometric resolution used in their model is not enough for the region, as well as

the exclusion of tide and river discharge in their analysis. Particularly, the tide plays a major contributing role in modulating the storm surge through tide-surge interaction (Krien et al., 2017b).

Given the discussed context, the objective of this paper is to estimate the risk of storm surge and associated flooding across the Bengal delta using a high-resolution, efficient, and well-validated regional coupled storm-surge model developed based on a higher-accuracy regional bathymetry (Khan et al., 2021a). To this extent, we integrated the wave-coupled hydrodynamic model

of Khan et al. (2021a) for a large ensemble (~3600 cyclones, ~5000 years of storm activity) of synthetic cyclones generated through the statistical-deterministic method, covering the whole range of natural variability of storm frequency, size, intensity and track location, with a dense spatial distribution. The interactions among the tide, surge, and waves are modelled explicitly at high spatial resolution. We then used the ensemble modelling results to study the storm surge-induced water level at various return periods, up to 500 years. In section 2, we describe our modelling framework, boundary conditions, and forcing generation

strategy, along with validation for recent cyclones. In section 3, the discussion continues with the probabilistic-deterministic cyclone ensemble. We show the results of the modelled storm surge-induced inundation mapping in section 4. A discussion is presented in section 5, followed by a conclusion in section 6.

## 2 Storm Surge Model

### 2.1 Model description

We developed our modelling framework based on SCHISM (Semi-implicit Cross-scale Hydroscience Integrated System Model). SCHISM is a derivative model by Zhang et al. (2016) built from the original SELFE (Zhang and Baptista, 2008). SCHISM uses an unstructured triangular mesh for spatial discretisation. SCHISM also includes a wetting and drying algorithm in the shallow areas. The use of semi-implicit schemes, in combination with the eulerian-lagrangian method for momentum advection,allows using large timesteps (typically, several minutes for an hectometric resolution grid) in SCHISM without compromising the

accuracy of the solution.

One of the consequential inputs in hydrodynamic models is accurate topography and bathymetry data. Over the Bengal delta, the publicly available datasets do not provide enough accurate data for reliable coastal modelling. We built our bathymetry

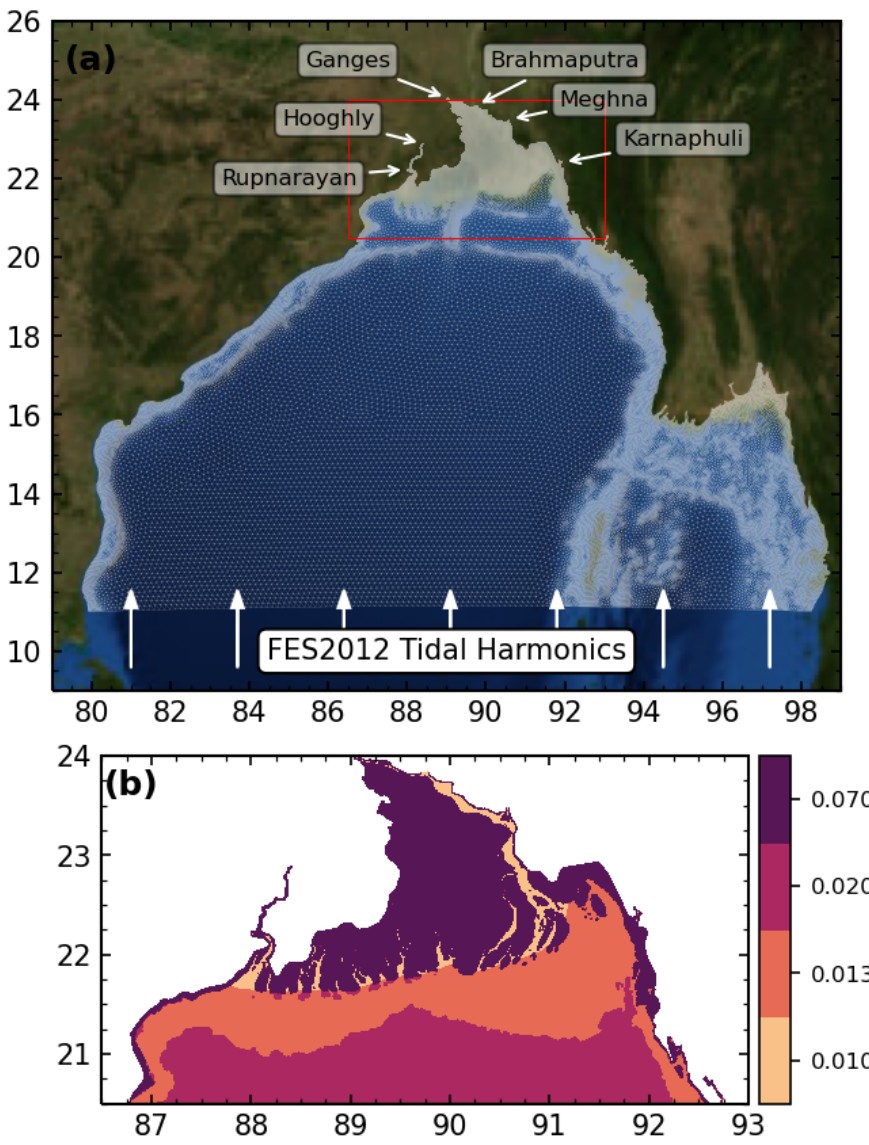

**Figure 2.** (a) Model domain with arrows indicating the open boundaries. (b) Distribution of Manning coefficient. The background used in (a) is taken from Blue Marble: Next Generation, credited to NASA Earth Observatory.

dataset on top of the work done by Krien et al. (2016). Their dataset is composed of a digitized nearshore bathymetry over the coastal region of Bangladesh and India, a dedicated river bathymetry from Bangladesh Water Development Board (BWDB),

and a high-resolution topography dataset over the Bangladesh part of the delta. We updated the bathymetry dataset of Krien et al. (2016) with an additional set of 34 new hydrographic charts collected from Bangladesh Navy, amounting to 77'000 sounding points (Khan et al., 2019).





We built our model mesh based on this bathymetry and topography dataset. The model domain covers the Bay of Bengal from 11°N to 24°N, comprising the whole delta and its surroundings (Figure 2a). We have used variable model resolution

based on the topography and wave propagation criterion, enhancing the resolution in shallow as well as sloppy areas. The final mesh resolution ranges from 15km over the deep parts of the ocean to 250m over the delta, amounting to about 0.6 million mesh nodes, and 1.1 million triangular elements. A time step of 300 seconds was found suitable for the resolution of our model (Khan et al., 2021a).

Hydrodynamically, our model is bounded on the ocean and the riverfronts, with the no-slip condition for momentum along

the closed boundaries. Along the ocean boundary, we have implemented a tidal water level derived from the 26 dominant tidal constituents of the global FES2012 tidal atlas (Carrère et al., 2013). Sensitivity tests between FES2012 and FES2014 (Lyard et al., 2021) did not yield any significant difference for our model domain. The applied constituents are - M2, M3, M4, M6, M8, Mf, Mm, MN4, MS4, Msf, Mu2, N2, Nu2, O1, P1, Q1, R2, S1, S2, S4, Ssa, T2, K2, K1, J1, and 2N2. In total, we implemented six open riverine boundaries. For the rivers Ganges, Brahmaputra, Meghna, Hooghly, Karnaphuli, we have

prescribed a discharge boundary condition. We have implemented a radiating boundary condition for the Rupnarayan river. To mimic the observed seasonal pattern of sea-level reported by Tazkia et al. (2017), we have implemented a harmonically-varying sea level along the southern boundary with a period of one year and an amplitude of 35cm, peaking in August. Over the whole domain of the model, the astronomical tidal potential is also forced for 12 constituents (2N2, K1, K2, M2, Mu2, N2, Nu2, O1, P1, Q1, S2, T2).

The bottom friction is parameterized using Manning's *n*. We adopted an approach for the spatial distribution of the Manning coefficient similar to Krien et al. (2016) and the one used in Khan et al. (2021a) (Figure 2b). For the deeper part of the ocean (depth>20m), we used a Manning coefficient value of 0.02. In the nearshore zone, the Manning coefficient is set to 0.013. In the rivers, a Manning coefficient value of 0.01 is set. Unlike in Krien et al. (2016), for the inland area, we adopted a Manning coefficient value of 0.07, which is more reasonable for a vegetated land like the Bengal delta (Bunya et al., 2010).

**2.2 Coupling with wave model**

Our modelling framework is online coupled with the spectral Wind Wave Model III (WWMIII) at the source code level (Roland et al., 2012). WWMIII solves the wave action equation over the same unstructured grid as the hydrodynamic core. It considers source terms for the energy input from wind (Ardhuin et al., 2010), non-linear interaction in deep and shallow water, energy dissipation due to white-capping, wave breaking, and bottom friction. We modelled the wave breaking using the

formulation of Battjes and Janssen (1978). A low value of the coefficient of breaking, $\alpha = 0.1$ (instead of $\alpha = 1$), is adopted to avoid over-dissipation over a very mild slope region like the Bengal delta (Pezerat et al., 2020, 2021; Khan et al., 2021a). We have discretized our modelled spectra into 12 directional and 12 frequency bins. For the current resolution of the model (e.g. 250m at the nearshore), we found that the sensitivity of the resulting wave-setup is negligible compared to higher resolution discretisation of the spectra (36 directional and 24 frequency bins). The coupled model is solved with implicit schemes, which

allows large coupling timesteps. In our model configuration, SCHISM and WWMIII were set up to exchange water level and current every 30 minutes.





## 2.3 Wind and pressure field

During a cyclone, the momentum transfer from the wind to the water column and the inverse barometer effect of the pressure drop are the primary driver of the storm surge. Thus, an accurate enough representation of the wind and pressure field in
a numerical modelling framework is of prime importance. In some well-monitored basins, there are extensive networks of observational data which can provide relatively accurate wind and pressure fields during a storm. However, the observations of these variables over the Bay of Bengal are virtually non-existent. As such, the wind and pressure field during a cyclonic storm have to be estimated from either numerical weather models or satellite observations. Due to the high error associated with the commonly available numerical weather models, it is a fairly common practice in storm surge modelling to infer the
wind and pressure fields from parametric cyclone models (Lin and Chavas, 2012). As the computation of parametric wind and pressure fields is comparatively light, this approach is also useful for sensitivity studies based on various configurations of a single storm (e.g. Hussain et al. (2017)) or from a large ensemble of synthetic cyclones (e.g. Krien et al. (2017a)).

The associated error with these parametric cyclone models varies with distance from the centre of the storms (Krien et al., 2018). The most considerable uncertainties in estimating wind and storm are from the surface wind reduction factor (SWRF)
and the choice of gradient wind profile (Lin and Chavas, 2012). From a comprehensive comparison of parametric cyclonic wind fields with scatterometer observations for 16 storms, Krien et al. (2018) showed that the relative error in the wind field minimises with a combination of parametric wind fields. Following their findings, we have used a combination of Emanuel and Rotunno (2011) and Holland (1980) formulations to derive the parametric wind profile. We have applied a SWRF of 0.9 (Lin and Chavas, 2012; Krien et al., 2017b). The effect of storm translational speed is accounted for through a surface background
wind reduction factor of 0.55 and a counter-clockwise rotation angle of 20° as suggested by Lin and Chavas (2012). In all cases, we used the Holland (1980) model to derive the pressure field.

## 2.4 Validation

The performance of our model in simulating the tide was documented in previous studies (Khan et al., 2019, 2020). We found that our model has a 2 to 4 times better reproduction of tidal water level compared to global tidal models, and comparable
performance with the previous state-of-the-art models in this region (Krien et al., 2016). The total complex error calculated over four major tidal constituents – M2, S2, K1, O1 – ranges from 5 to 20cm, which is comparable to other well-documented shorelines around the world. Notably, the current model shows better performance around the mouth of Meghna estuary – a critical region with complex hydro-morpho-dynamics.

The observational network of tide gauges is sparsely distributed, and typically non-functional during an intense cyclone
(Krien et al., 2017b). It is thus relatively difficult to achieve a thorough assessment of the model realism for most of the historic cyclones. The quality of the tide gives a preliminary assessment of the potential skills of our model in simulating storm surges. Given the wind and pressure fields are adequately prescribed, and necessary physics – particularly the wave coupling – is incorporated, the model is expected to simulate a storm surge appropriately.



In Khan et al. (2021a), the performance of the model in simulating storm surges is demonstrated through a hindcast of the
surge generated by a recent cyclone – Amphan – that made landfall in this region. From a set of in-situ observations, it was
found that the model performs well (5-10 cm peak water level error) in reproducing the maximum water level observed at
multiple tide-gauge locations distributed throughout the Bengal shoreline.

Additionally, similar to Krien et al. (2017b), we also performed a hindcast of the water level generated by cyclone SIDR –
another comparatively well-documented historic cyclone. Figure 3 shows the water level evolution at four tide gauge stations
around the landfall location. The modelled water level is shown in black lines, and the observed water level records are shown
in dots. As noted by Krien et al. (2017b), the tide gauge stopped working at Khepupara at the time of very high water. The error
of the peak modelled water level typically amounts to 10-20 cm. It is to be noted that, in their hindcast, Krien et al. (2017b)
shifted the time of landfall by half-an-hour to get a better match of the tidal propagation. Here we retained the original JTWC
track hence the slight shift in phase of the water level. However, the important thing to note here is the peak water levels are
well captured, for both Amphan (Khan et al., 2021a) and Sidr (Figure 3). This peak water level is the main variable we will
deal with in our storm surge hazard assessment.

The results of these experiments – cyclone Amphan (Khan et al., 2021a) and Sidr (this section) – suggest that our model
can successfully reproduce water level evolution, as well as maximum water level, during cyclones along the shoreline of the
northern Bay of Bengal.

In addition to the validation experiment described above, we performed a hindcast for the cyclones shown in Figure 1. The
corresponding surges and the estimated maximum water level estimate from our model are listed in Table 1. We adopted JTWC
best track dataset for the cyclones track and intensity information. (For the 1970 Bhola cyclone, the intensity of the cyclone
was not available in the JTWC dataset.) The motivation for doing so is not validation. As illustrated by (Krien et al., 2017b)
validating for past cyclones is often difficult due to the limited availability of water level data, which is further aggravated by
the failure of the tide gauge during cyclonic storms. The objective is rather to illustrate the typical variation of maximum water
level that can occur during a cyclone along the Bengal delta shoreline.

## 3   Probabilistic-Deterministic Cyclone Ensemble

### 3.1   Synthetic storm dataset

Here we use a large ensemble of 3600 synthetic storms passing by the lower part of the Bengal delta. These synthetic storms
were generated through the statistical-deterministic approach of Emanuel et al. (2006) based on the climatic conditions of the
1981-2015 period. The forcing background climate is derived from the National Center for Environmental Prediction/National
Center for Atmospheric Research (NCEP/NCAR) reanalysis (Kalnay et al., 1996). In this method, synthetic storms are gener-
ated in three steps. First, a synthetic environmental wind field is derived from a long-term dataset which conforms to the mean
climatology. For our ensemble of storms, the environmental wind field is derived from the NCEP/NCAR reanalysis dataset
for the zonal and meridional wind components at 250 hPa and 850 hPa. During this step, the observed monthly means, and
variances are respected, as well as most of the covariances. Monthly means of sea surface temperature and atmospheric tem-




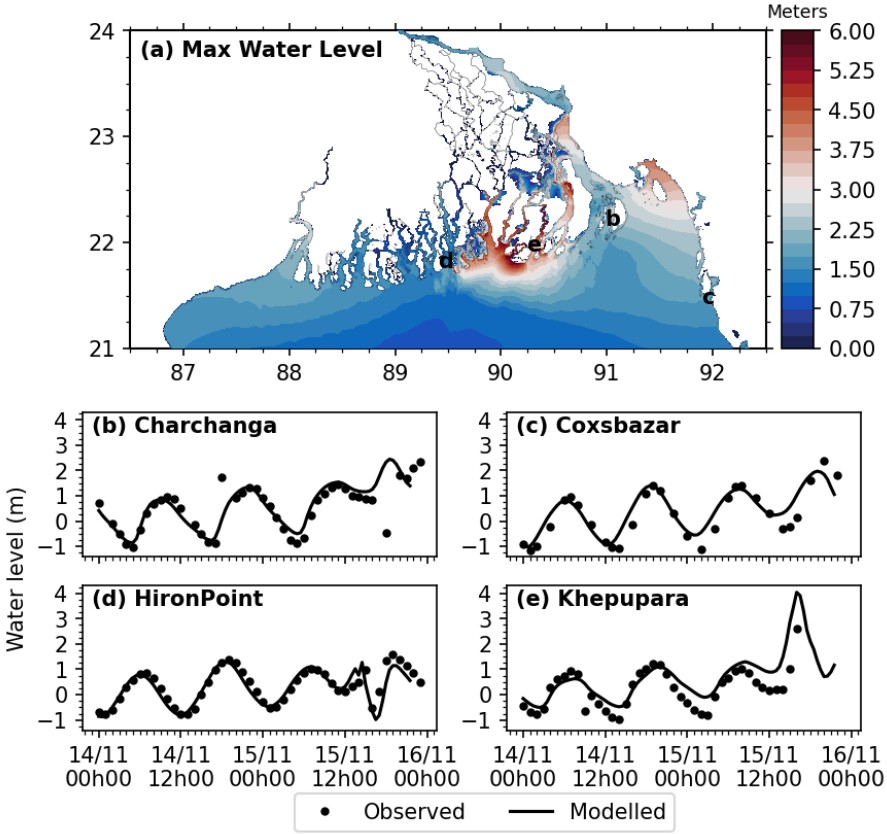

**Figure 3.** (a) Maximum simulated water level during cyclone Sidr. Comparison of simulated and observed water level during cyclone Sidr at (b) Charchanga, (c) Coxsbazar, (d) Hironpoint, and (e) Khepupara. The station locations are shown in (a).

**Table 1.** Simulated maximum water level for the cyclones shown in Figure 1. $V_{max}$ is the maximum wind speed in knots, $WL_{max}$ (Sim) is the simulated maximum water level range over the delta shoreline in meters.

| No | Cyclone | Year | Casualty | $V_{max}$ | $WL_{max}$ (Sim) |
|----|---------|------|----------|-----------|------------------|
| 1 | Bhola | 1970 | 200-500k | 100 | NA |
| 2 | Gorky | 1991 | 64-139k | 140 | 5-8 |
| 3 | Sidr | 2007 | 3.5k | 140 | 4-5 |
| 4 | Aila | 2009 | 339 | 65 | 3-4 |
| 5 | Ruanu | 2016 | 227 | 60 | 4-6 |
| 6 | Mora | 2017 | 135 | 80 | 2.5-3.5 |
| 7 | Fani | 2019 | 89 | 135 | 2.5-4.0 |
| 8 | Bulbul | 2019 | 41 | 85 | 3.5-4.5 |
| 9 | Amphan | 2020 | 118 | 140 | 3-5 |





perature and humidity are also used. In the second step, a set of tracks is generated from a random field of genesis points and a weighted mean of 250 hPa and 850 hPa wind plus a correction for beta drift (Emanuel et al., 2006). The weight factor is tuned to best match observations of track displacements, and the rate of seeding is calibrated to match overall observed storm

frequencies. Finally, the storm intensity is estimated along the tracks using a coupled ocean-atmosphere one-dimensional axisymmetric model with parameterized effects of vertical wind shear (Emanuel et al., 2004). The wind shear is given by the synthetic time series of winds determined previously. The output of this axisymmetric model is essentially the parameters of the radial structure of the pressure and wind field, expressed as along-track series of central pressure, maximum wind speed and the radial distance from the centre of the storm where this maximum wind speed prevails. Previous applications of this

model include the assessment of storm surge hazard in Guadeloupe (Krien et al., 2015) and Martinique (Krien et al., 2017a), storm surge return period in New York City (Lin et al., 2010), as well as the estimation of wind return period in Boston and Miami (Emanuel et al., 2006).

Given that our geographic interest is focused on the Bengal delta, the cyclone ensemble comprises only the cyclones that pass through a 300 km radius centred on the centre of the delta (defined at 89.57°E, 21.71°N) . With an average annual frequency

of 0.70314 cyclone, the ensemble of 3600 cyclones considered here represents more than 5000 years of cyclonic activity over the northern Bay of Bengal, under present climate conditions. The landfalling cyclone frequency along the shoreline (for each 20km × 20km pixel) is shown in Figure 4(a). The inset of Figure 4(a) shows a subset of the cyclone tracks in the ensemble.

This synthetic ensemble of storms is fundamentally different from the hypothetical storms employed in past studies, e.g. Hussain et al. (2017). Hypothetical storms are typically generated from a known storm by modifying its track or strength or

both, which could render some storms unrealistic. On the other hand, the storms in our ensemble are physically meaningful and statistically consistent. The consistency of the cyclone ensemble is illustrated in Figure 4 (b) and (c). The distribution of the simulated maximum wind speed ($V_{max}$) shows a good agreement with the observations from JTWC (Figure 4b).

Similarly, the seasonal distribution of the cyclone ensemble and that observed from JTWC both show a well matching pattern with a bimodal seasonality (Figure 4c) (Alam and Dominey-Howes, 2014). In the Bay of Bengal, low-pressure systems

typically cannot intensify into a storm due to strong vertical wind shear present during monsoon (June-August). During the pre-monsoon (March-May) and post-monsoon (September-November), low vertical wind shear, and high sea surface temperature provide a suitable condition for low-pressure systems to intensify. The simulated bimodal temporal evolution of the synthetic cyclone indicates that the statistics captured by the statistical-deterministic method correspond well with the seasonal climatic characteristics.

**3.2 Ensemble surge estimate**

Estimation of the extreme water levels associated with the storms in the ensemble is done in two steps. First, for each synthetic storm, the wind and pressure fields are derived, as explained in Section 2.3. Here, for the inner core of the cyclones ($r < r_{50}$), the wind field is estimated using the model of Emanuel and Rotunno (2011). On the outer core ($r >= r_{50}$), we used the Holland (1980) model. The pressure field is estimated from the formulation of Holland (1980) for all radial distances. The wind and

pressure fields are updated every 15 minutes with a linear interpolation of the associated variables.



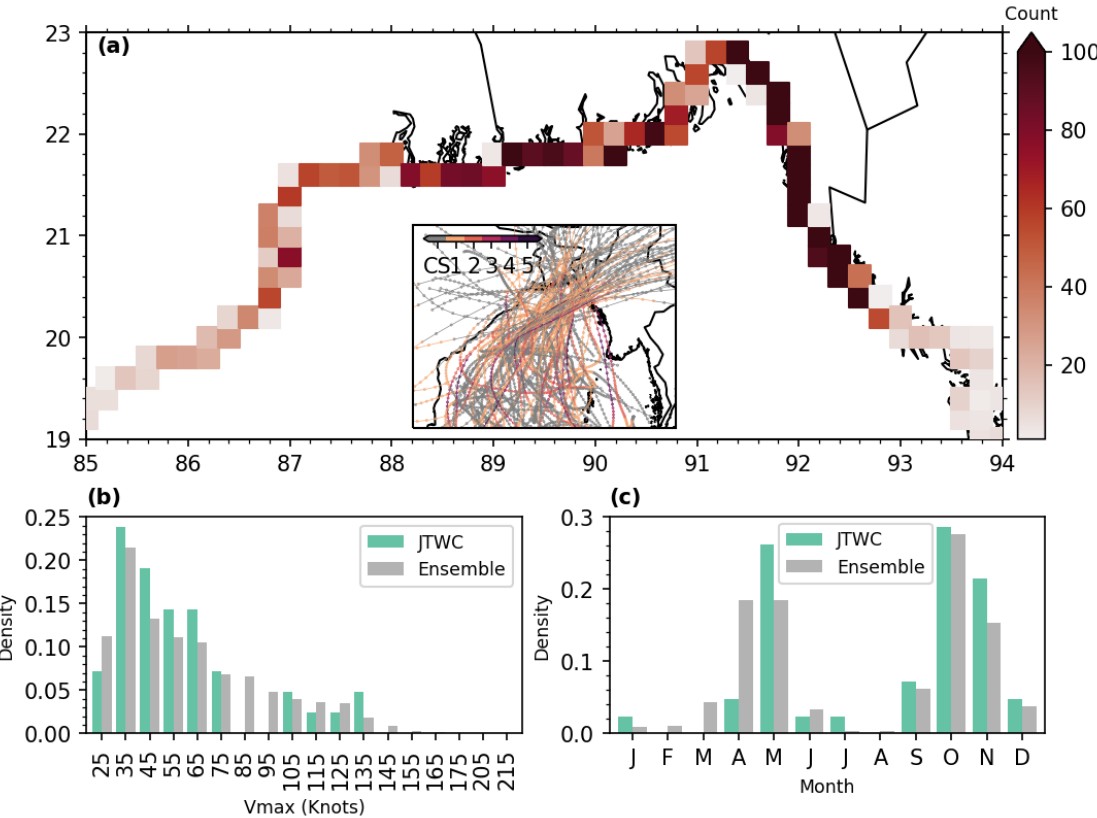

**Figure 4.** a) Spatial distribution of the paths of the cyclones that make landfall along the coast of Bengal delta. Each square bin is 20 km wide. A small subset of cyclones trajectories is shown in the inset. (b) Distribution of maximum wind speed of the synthetic cyclones compared to the JTWC dataset, (c) Annual distribution of the occurrence of the synthetic cyclones compared to JTWC dataset.

Second, with wind and pressure fields, we calculate the ensemble of water level estimates using the same ocean-wave coupled model set-up used for the hindcasts described in the previous section. We achieved this ensemble of water level estimates through individual simulation of each of the 3600 storms – which amounts to 14 years worth of model simulation. Along the open boundaries of the rivers Ganges, Brahmaputra, and Meghna, we have prescribed a time series of observed discharge sampled at the timestamp of the cyclones, which ranges from 1980 to 2015. We have applied a discharge climatology as the boundary condition for Hooghly (Mukhopadhyay et al., 2006), and Karnaphuli (Chowdhury and Al Rahim, 2012). Similar to the model set-up used for the hindcast experiments, we have also taken into account the seasonal mean sea level variation of the Bay of Bengal by applying an annual harmonic on the water level along the oceanic open boundary with a period of 1-year peaking in August with an amplitude of 35cm, as in Tazkia et al. (2017). The coupling between SCHISM and WWMIII allows accounting for the wave set-up induced by the radiation stress gradient of the waves (Longuet-Higgins and Stewart, 1964). Similar to the model set-up for cyclone Amphan hindcast in Khan et al. (2021a) and for the Sidr hindcast in this paper, the





breaking coefficient $\alpha$ is set to 0.1 (instead of the default parameter $\alpha = 1$) to avoid over dissipation of the waves over the submarine part of the delta with mild slopes (Pezerat et al., 2021; Khan et al., 2021a). We also invoked the wetting and drying algorithm in SCHISM with a threshold of 10cm depth of water for an element to be registered as wet.

## 3.3  Statistical analysis methodology

We have applied ranking-based statistical analysis to infer the quantities at a given return period. To do so, at each node the quantity of interest (e.g., maximum water level, maximum surge level) is first sorted sequentially among the 3600 members, from the smallest value to the largest one. As we mentioned, since our dataset consists of 3600 cyclones with a yearly frequency of 0.70314 cyclones, it corresponds to 5120 years of cyclonic activity. Once sorted in ascending order at each point of the model
grid, each water level present in the sorted ensemble corresponds to a return period ranging from 5120/3600 viz. 1.42 years (for the smallest return period) to 5120 years (for the largest return period). Estimated quantities at return periods between 5 years and 500 years (at 5-year steps) are considered trustworthy in our subsequent analysis, to make sure that our ensemble contains a large enough sample for each return level. Over the whole Bengal delta, at 500 years level, the number of individual unique cyclones that contribute amounts to about 50 (viz. an average of three cyclones per 20km of shoreline). Above 500 years, the
sample size for cyclones that contributes to the return period is smaller, which is potentially too small to capture the extreme value statistics. We linearly interpolated the water level at individual integer return periods (in years) from the ranked dataset.

## 4  Storm Surge Hazard

The outcome of the statistical analysis consists of water level maps at various return periods, spatially distributed on our model grid. Figure 5a shows the water level at the 50-year return period over the northern Bay of Bengal. The meaning of the colour
bar is twofold. At all grid points located below mean sea level, including the estuaries the colour indicates the water level above the mean sea level. At the grid points located above mean sea level (e.g., inland), the colour indicates the inundation depth above topography. The 50-year return period water level shows a similar spatial pattern as the tidal range (Khan et al., 2020). We see that a dipolar pattern arises, with a very high-water level in the north-eastern (around 88°E) and the north-western (around 91.5°E) corners of the bay. In contrast, in the central region (between 88.5°E and 91°E), there is a local minimum of about
2.25m-2.5m in the estimated 50-year return period water level. However, among these two maxima, the 50-year water level in the north-eastern part reaches about 6m, twice as much as the 50-year water level in the north-western part of the delta (barely 3m).

Inland, at a 50-year return period, a contrasting flooding pattern is observed. Two specific regions are found to be flooded by a storm surge at 50-year return period - one around 89-90°E 21.5-22.5°N, and another around 90-91°E 22.5-23.5°N. The first
inundated region is the Sundarbans mangrove forest, which spans across the border between Bangladesh and India. The region is a sanctuary and legally protected region, and thus devoid of any engineering intervention, like embankments. The second inundated region along the GBM estuary, in contrast, is densely populated. According to the polder embankments dataset used in our modelling, this region is not protected by the coastal polder network and thus exposed to the risk of flooding. Other than

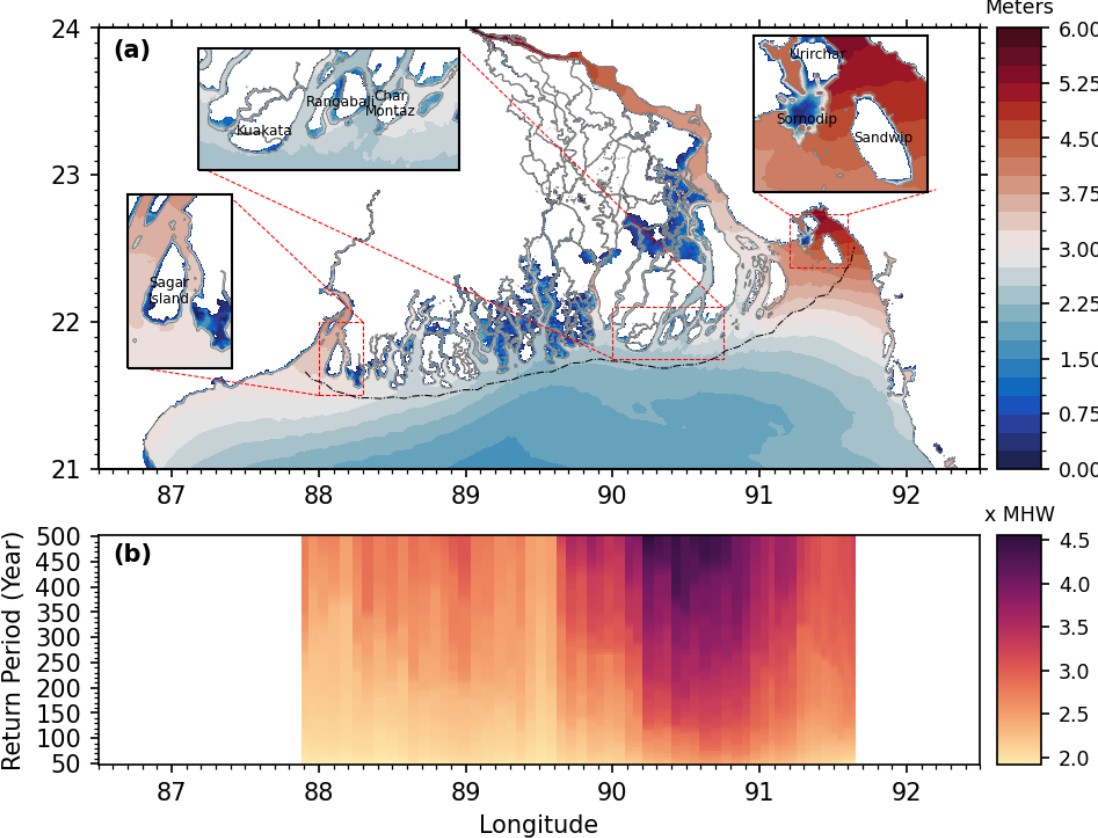

**Figure 5.** (a) Inundation extent and corresponding water level at 50-year return period. (b) Water level for the 50-500 years return period expressed as a multiple of the yearly mean tidal high-water level along the nearshore dash-dotted line shown in (a).

these two regions, the sea-facing inland areas are primarily found to be not flooded at a 50-year return level. These regions

are protected by polder embankments (typically earthen) (Khan et al., 2021a), hence creating this contrast in the pattern of the flooded area.

Along the delta coastline facing the open ocean, the 50-year return period water level appears contrasted, as we saw. To further illustrate this, we extracted the water level at the various return periods along the black line shown in Figure 5a, and we display it in terms of multiple of the mean high-water level (MHW) along the line (Figure 5b). The MHW level is calculated

from a 1-year long tidal simulation, and taking the mean of daily (25-hour) tidal high-water estimate. Our motivation for such a display stems from the similarly contrasting range of MHW along the shoreline – with two macrotidal poles in the north-western and north-eastern corners reaching around 2m (at 88°E) to 3m (at 91.6°E) and a dip of 1 to 1.25m around 90°E to 91°E. From Figure 5b, it can be seen that throughout the cross-section, the 50-year return period water level is around twice as much as MHW or higher. The increase in water level with the return periods along this nearshore cross-section shows regional



sensitivity. From the mouth of Hooghly estuary to the region of the Sundarbans mangrove forest (88°E-89.5°E) the water level evolves moderately with return period showing only 50% increase from 50-year return period to 500-year return period. On the other hand, when approaching the mouths of the Meghna estuary, between 90.5°E and 91°E, the pluri-centennial water level increases sharply with the return period, to reach more than twice the 50-year return period water level for the 500-year return period. The maps of water level over the delta obtained for return periods ranging from 5 years to 500 years are shown in

Figure 6. The regional pattern of water level at various return periods consistently shows the same bi-polar pattern throughout the return periods investigated here, from 5-year to 500-year return periods (Figure 6).

The inundations in the mangrove region around 89.5°E seem to have a large saturation effect on the evolution of the return period of the water level. There the water level remains practically the same for the various return periods, slightly rising from 2.5m at 50-year return period to 3m at 500-year return period. We also see this evolution in Figure 6.

The flooding extent in the estuaries and adjoining areas changes significantly from the short return periods to the longest ones (Figure 6). We selected the three main estuaries to take a closer look at the evolution of water level with the return periods. These are Ganges-Brahmaputra-Meghna (GBM) on the eastern side (Figure 7a-b), Pussur estuary in the central mangrove region (Figure 7c-d), and Hooghly estuary on the western side (Figure 7e-f). The water level at the various return periods is extracted along the lines shown in Figure 7a,c,e and shown as a multiple of the 50-year return period water level in Figure

7b,d,f.

In the GBM estuary, the water level increases sharply with increasing return period along the whole cross-section considered here, as shown in Figure 7a-b. Over the downstream-most 50km, the 500-year return level reaches more than twice the 50-year return level. Further upstream, up to the estuarine bottleneck of Chandpur (23.24°N), the sensitivity of the water level to the return period is more moderate, with around 50% higher water level at 500-year return period compared to the 50-year return

level.

In the Hooghly and Pussur estuaries, a contrasting pattern is observed compared to GBM – the water level for the longer return periods rises much more sharply in the upstream part than in the downstream part. Along the Hooghly estuary, the sensitivity of the water level to the return period is moderate for the first 100km but amplifies considerably further upstream. Along the Pussur estuary, weak variation of the water level across the various return periods is seen over the first 25km and

then shows a moderate evolution towards upstream.

## 5   Discussion

### 5.1   Synthetic cyclone ensembles statistics

The synthetic cyclone dataset used in this study makes our estimation inherently different from most of the previous attempts to map the storm surge hazard along the shoreline of the Bengal delta. The underlying statistical-deterministic approach of

our storm database provides three significant improvements over the previous approaches used over the Bengal delta. First, the random genesis of the storms induces a probabilistic nature of the population of the storms. Second, the deterministic evolution of the storms makes sure that the resulting population of tracks are physically valid, consistent with the ocean-atmosphere


**Figure 6.** Inundation extent and corresponding water level at (a) 5-year, (b) 10-year, (c) 20-year, (d) 50-year, (e) 75-year, (f) 100-year, (g) 250-year, (h) 500-year return period.



**Figure 7.** Water level as a function of the return period, expressed in multiple of the 50-year return period water level, along the dashed lines shown in red for (a)-(b) Ganges-Brahmaputra-Meghna estuary, (c)-(d) Pussur estuary, (e)-(f) Hooghly estuary. The x-axis is the distance from the mouth of the estuary, in km. The background shading in panels (a), (c) and (e) shows the bathymetry in m.





coupling to a reasonable extent. The resulting statistical distribution of maximum wind speed of the storms shown in Figure 4b gives confidence in the capability of the statistical-deterministic approach to capture the actual behaviour of the ocean-
atmosphere system over the Bay of Bengal. Third, a large enough population of storms allows both an appropriate spatial coverage of the coastline and appropriate temporal coverage of the pluri-annual to pluri-centennial return periods of interest here (Figure 4a,c).

The distribution of the cyclones along the shoreline shown in Figure 4a illustrates a dense population of cyclones making landfall along the convex delta front (88°E-91°E). An affluent population of storms is also observed along the eastern coast of
Bangladesh. This landfall pattern corresponds to previous observations that the landfalling cyclones in the Bangladesh coastline tend to move north-eastward (Ali, 1996).

## 5.2  Inundation extent and consistency

The storm surge inundation hazard has significant spatial variability across the Bengal delta region (Figure 5). Our underlying statistical analysis is an empirical rank-based return period estimation, which requires a large enough size of the ensemble
to achieve a smooth spatial and temporal resolution. Visibly, the water level estimation at the 50-year return period shown in Figure 5 follows a smooth pattern over the Bengal delta region. In our estimate, about 600 and 400 unique cyclones contribute to the 50- and 100-year return period water level respectively. In other words, for a given return period the associated water level is pulled-in from a large number of individual cyclones, which in turn includes cyclones with a range of intensity landfalling at various phases of the tide. Indeed, this spatial smoothness of water level is observed for all the return periods we analysed
from 5 to 500 years (Figure 5), indicating a robust estimate of the storm surge water level.

It appears that the water level has a similar regional distribution as tidal amplitude with two maxima in the north-eastern and north-western corners, and a relative minimum in-between (Figure 5a). At the mouth of Hooghly, in the north-western corner of the delta, the 50-year return period water level is about 3.5m. The 50-year water level reduces to about 2m when moving eastward at the mouth of the Pussur river. Further eastward the water level increases again to reach about 6m along the
shoreline of Sandwip Island (91.6°E, 22.5°N) and Chittagong (91.7°E, 22.5°N). This estimated water level is on average twice as much as the mean high-water level (Figure 5b). We found the highest 50-year return period water level at the eastern side of the mouth of Meghna estuary (90°E - 91°E). There the 50-year return period water level reaches thrice as much as the mean high water.

## 5.3  Comparison with previous estimates

As mentioned previously, a reliable estimate of the return period of storm surge water level over the Bengal delta has not emerged due to the gap in data as well as modelling limitations. A comparison is presented here with the previous estimates by Lee (2013) and Jakobsen et al. (2006). Lee (2013) did an extreme value analysis of the observed time series over 1977-2009 (33 years), at Hiron Point tide gauge located in the central Bengal shoreline (89.5°E, 21.8°N). For this purpose, he decomposed the time series using the Ensemble Empirical Mode Decomposition approach. He reconstructed the water level by only keeping
the very high-frequency (~3 hours) and very low-frequency modes. This process essentially removed the tide from the water





level and retained the detided residual (i.e., surge). Using a yearly-maximum method, in his extreme value analysis, he obtained a 1.66m surge level at 50-year return period. In the previous section, our analysis was focused on the water level rather than surge level. To compare with the estimate of Lee (2013), we reprocessed the whole ensemble of storm event simulation results. We have first extracted the tidal water level from the 3600 cyclones that we simulated. Then, for each cyclone, we extracted
the maximum surge level. Finally, on this maximum surge estimate, we applied the same ranking based return period estimate. At 50-year return period, our estimation of surge amounts to 1.8m at Hiron point. With a difference of only 14cm (1.66m estimated by Lee (2013)), our estimated value is comparable to the estimated value by Lee (2013) from the observation time series. To be noted that, the estimated 50-year return period water level from our ensemble is about 3m at Hiron point.

The evolution pattern of 50-year return period water level is consistent with the previous estimates but with varying degree
of agreement on the amplitude (Jakobsen et al., 2006; Sindhu and Unnikrishnan, 2011). From an ensemble of 17 historic cyclones, Jakobsen et al. (2006) estimate that the 100-year return period water level reaches about 5m at the mouth of Meghna, and about 8-10m around Sandwip (See their Figure 5). A similar pattern has emerged in our analysis but with a much different water level estimate. We estimate that the 100-year return period water level is about 4m at the mouth of Meghna, reaching about 6m around Sandwip. In general, the water level estimates of Jakobsen et al. (2006) are consistently larger than the one
presented here. It is noteworthy that the modelling framework of Jakobsen et al. (2006) does not take into account the wave set-up. However, the limited and potentially biased sampling of the "strongest" cyclones (17 in total, over 40 years) leads to an overestimation of the storm surge level.

Along the eastern part of the West Bengal shoreline (87°E-88°E), Sindhu and Unnikrishnan (2011) found an increase in water level when moving northward. Similar to Jakobsen et al. (2006), the pattern of water level they estimated is similar to
ours, but their values are significantly larger than ours. At Sagar Island, their estimated 5- and 50-year return period water level is 6.92m, and 8.74m respectively. In our estimation, the 5- and 50-year return period water level is 2.75m and 3.5m above mean sea level respectively. It is not clear if the estimate of water level estimated by Sindhu and Unnikrishnan (2011) is relative to mean sea level. It is to be noted that the mean estimated high water at Sagar Island is about 1.75m above mean sea level which is only one-fourth of the estimated 5-year return period water level (6.92m) by Sindhu and Unnikrishnan (2011).

Leijnse et al. (2021) used a somewhat similar approach to ours. They used an extreme value analysis on the surge estimate of 1000 year simulated cyclonic activity and estimated that at Charchanga and Chittagong the surge level at 100 (10) year return period is about 2.8 (1.4) m and 3.4 (1.6) m respectively. To better compare with the estimate of Leijnse et al. (2021) we have re-simulated the ensemble of our 5000 year cyclonic activity (about 3600 cyclones) again, but without incorporating the tide - e.g. only the storm surges. In our estimate the 100-year return period surge level is 3.6m and 4.1 m for Charchanga and
Chittagong respectively. In other words, in these two locations, the estimate of Leijnse et al. (2021) is more than 60cm lower than ours.

Several factors might underlie this difference between the estimate by Leijnse et al. (2021) and ours. As discussed before, the modelling framework of Leijnse et al. (2021) is an extraction of a global model relying on global bathymetric data which suffers from large bias in the northern Bay of Bengal region (Krien et al., 2016). The model resolution used in Leijnse et al.
(2021) is also kilometric (3km or higher for the unstructured-grid hydrodynamic model, 2km for the structured-grid wave





model), which is not enough to model the complex shoreline and shallow coastal domain of the northern Bay of Bengal, particularly the wave breaking and associated setup. The parameterization of the wave breaking model might also not be well tuned for the very mild slope of this region (inferior to 1/5000). As shown by Pezerat et al. (2021), the default parameterization in wave breaking models can induce over-dissipation over such mild-slope regions and underestimate the wave setup by as

much as 100%. Additionally, no discharge is imposed along the Ganges-Brahmaputra where the surge can interact with the river discharge and propagate far inland (Khan et al., 2021a). Finally, Leijnse et al. (2021) used extreme value analysis based on Peak-over-threshold (POT) and exponential distribution fitting to infer the return period surge level, which could induce further error linked with the choice and fitting of the distribution, and the threshold for POT method.

### 5.4   Return period of previous cyclones

From the hindcasts of previous cyclones – cyclone Sidr and Amphan – a wide regional variation of the return period of the maximum water level during storm surge is found.

    The maximum water level estimates from our hindcast of the Sidr cyclone (see Figure 3) shows a strong surge near the landfall location where it reaches 6m along the shoreline of Kuakata and remains strong in the connected tributaries. This water level amounts to a return period of about 500 years. At the same time, the maximum water level hindcast along the coast

of Chittagong and Sandwip is about 4.5m, which corresponds to roughly 50-year return period water level at that location (Figure 6).

    Compared to Sidr, the regional variation of maximum water level is much less pronounced for cyclone Amphan (Khan et al., 2021a) (See their Figure 5). The maximum modelled water level reached about 5m around (88.4°E), which corresponds to a 250-year return period. At the landfall location of Sidr (around Kuakata), about 100km eastward, the maximum water level is

estimated to be 2.5m, corresponding to 50-year return level. Similar to Sidr, the water level along the Sandwip coast during Amphan reached about 4.5m, i.e., 50-year return level.

### 5.5   Role of embankments

From the various maps of n-year return period inundation extent and water level shown in Figure 6, it is clear that the embankments play a vital role in the storm surge flooding. Although these embankments were established for controlling tidal intrusion

to promote agricultural activity, the initial designed heights appear to be relevant, providing considerable protection during extreme events. Our estimate shows that, under current climate cyclonic activity, for the embankment heights implemented in our model, the embankments start overflowing at 75 to 100-year return period water level (Figure 6).

    The level of embankment protection estimated here must be taken with caution as we are limited by the availability of the embankment information and up-to-date status report. These earthen embankments are poorly maintained (Islam et al.,

2011), whereas our modeling approach essentially assumes that their geometry is constant in time (implying that they can not be altered by any devastating cyclone typically). This is not the case as shown by Khan et al. (2021a) in their newspaper survey, where they found the breaching of the embankment to be the prevalent cause of flooding inside a polder during cyclone Amphan. The information of embankments' height used in the model of this study is more than a decade old. The actual




current height of these embankments is unknown. Additionally, in our embankments height dataset, only one value of height is
provided across each of the embankments. In reality, we expect a spatial variation of embankments height. Considering all these
factors, the estimated level of return period when the embankments start overflowing might be overestimated. In other words,
the embankments are estimated to be overflowed at a higher return period water level event, where in reality it is overflowing
at a lower return period.

## 5.6 Population exposure

Storm surge is a significant hazard for the large population that lives in the Bengal delta. Figure 8 shows the population count
per pixel in Bangladesh part of the delta area with a grey colour bar based on GPWv4 dataset (Center For International Earth
Science Information Network (CIESIN), Columbia University, 2016). The contours of the flooding extent at return period
ranging from 5-year to 500-year are shown in colour.

The estimation of the population living under a given return period of water level is not straightforward. The GPWv4 dataset
generally resolves the population at the second administrative level (Upazila), which typically covers hundreds of square
kilometres of area. However, the data itself is provided at a regular longitude-latitude grid. For a given administrative area, a
constant density is assigned for all the pixels that fall under the administrative boundary. On the other hand, the model grid is
unstructured, and the resolution of our model output is sub-kilometric. Hence it is first necessary to map the inundation over
the population dataset. We took the following steps for mapping :

1. We interpolated the modelled return period water levels to the same regular longitude-latitude grid as the population
dataset (30" resolution, about 1km over our region). As the model grid is triangular, we used a linear triangulation
interpolation method. We also interpolated the bathymetry onto the population grid in a similar fashion. After the water
level interpolation, we considered a pixel inundated if the inundation level is 10cm above the topography. A version of
the interpolated return period water level dataset (Khan et al., 2021b) is made publicly available to accompany this article
(see Code and data availability).

2. We set the pixels in the population dataset associated with topography below mean sea level in the model to null. The
pixels outside the model boundary are also set to a null value.

3. For each return period, we counted the exposed population if the pixel is registered as inundated in step 1.

The estimated population living in our model domain over the Bengal delta extent shown in Figure 8a amounts to 30
million. This count amounts to the fraction of Bangladesh population living at an elevation 5m or less. The density of the
exposed population at various return periods of inundation is shown in Figure 8b. Our estimate shows that about 2 million
people currently live within the 5-year return period flood level area. Even if the embankments were to work without failure
during a cyclone, about 3 million more people would get exposed to the flooding of 50-year return period. This additional
count of the population represents about 10% of the total population living inside the study area. At a 100-year return level, the
number of exposed people increases to about 21% of the total population inside the modelled domain.

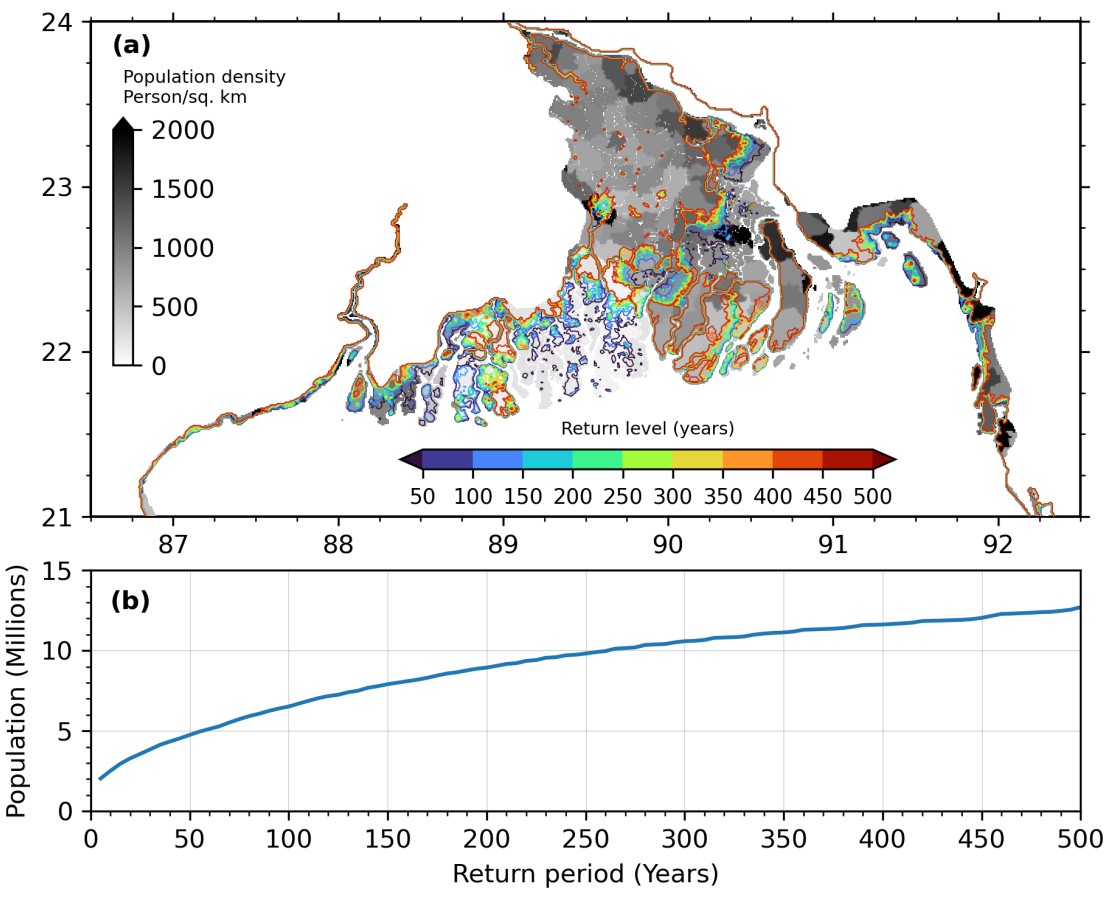

**Figure 8.** (a) Spatial coverage of flooding at various n-year return period estimated from the model. Gray colorbar shows the population density per square kilometre. (b) Number of people affected at various n-year water level return period.

A refined estimation of the exposed population requires further refinement of the population spatial distribution, which is out of the scope of this work, and not pursued further. In this assessment, we did not consider the probable (but not publicly documented) existence of city protection embankments at local scale, which may distort the patterns of Figure 6a locally. We did not consider either the potential degradation of the dikes, and possible dike breaching during an intense cyclonic event.

Knowledge of these factors will surely impact the anticipated exposure of the population to flooding suggested by our analysis. Instead, our focus was mainly on the physical mechanism of the flooding from storm surges. Our exposed population map, although preliminary, provides useful and spatially continuous information at relevant spatial scales to document the exposure to storm surge flooding and to better understand the vulnerability of the vast population of the Bengal delta continuum.


## 6   Conclusions and Perspectives

In this study, we present a robust estimate of the return period of water level over the whole Bengal delta shoreline under
current climate conditions. We based our estimates on a large ensemble of statistically and physically consistent cyclones and
high resolution coupled hydrodynamic modelling. For the first time, we estimated the extreme water levels associated with
storm surge for a broad range of return periods from 5 to 500 years, at sub-kilometric spatial resolution over the Bengal delta.
We reported a complex pattern of water level extreme values both along the shorelines and within the estuaries. Compared to

previous studies, and despite our comprehensive modeling framework explicitly accounting for the various components of the
storm surges (including the waves), we concluded a water level at a given return period lower than most of the past studies –
e.g. Jakobsen et al. (2006). On the other hand, we concluded a higher surge level compared to the past studies – e.g. Leijnse
et al. (2021). We show that embankments play a significant role in determining the flooding pattern. Our estimate suggests
that at least 10% of the coastal population is currently exposed to storm surge inundation at 50-year return period. The return

period water level derived here could provide valuable information for robust engineering, social, economic assessment and
future policy decisions. The maps of water level extremes we worked out should also be a valuable basis for zoning the risky
areas, and should favour an efficient resource allocation for pre-cyclone preparedness. The diverse range of extreme water level
should also nourish future research directions to better understand the dynamics of extreme water level over the continuum
of the Bengal delta. Evidently, continued future research is necessary under the threat of climate change with unavoidable

sea level rise (Oppenheimer et al., 2019; IPCC, 2021), land subsidence and morphological changes (Paszkowski et al., 2021),
and potential increase in the frequency of devastating cyclones under future climate (Emanuel, 2021). We acknowledge the
potential compounding effect of rainfall during the storm on inland flooding which is not considered in our analysis. While
our modelling framework is capable of incorporating rainfall, the mechanisms of pluvial flooding during cyclonic storms are
not yet established in this region and they are expected to depend strongly on the topography, embankments, dense network of

road and hydraulic structures in place. We also remain humble about the potential drawback in our model set-up resulting from
our inaccurate knowledge of the actual height of the embankments and potential existence of city protection embankments
(Khan et al., 2021a), so the upstream flooding extent seen in Figure 6 may be considered with caution. One may revisit these
questions once a more reliable and consolidated database of bathymetry and topography (especially dikes heights) becomes
available across the delta region.

*Code and data availability.*   The instructions to download and install the model used in this study can be accessed freely at https://github.
com/schism-dev/schism. The estimated storm surge water level at a return period ranging from 25 to 500 years in a 30" regular lat-lon grid
can be found at https://doi.org/10.5281/zenodo.5614101(Khan et al., 2021b).

*Author contributions.*   JK,FD,YK designed the study. KE provided the storm ensemble. JK did the modelling, analysis, illustrations, and
wrote the first draft. All co-authors contributed to editing.





*Competing interests.* The authors declare that they have no known competing financial interests or personal relationships that could have appeared to influence the work reported in this paper.

*Acknowledgement.* We acknowledge Joint Typhoon Warning Center (JTWC) for providing the observed JTWC dataset. This work was granted access to the HPC resources of IDRIS under the allocation made by GENCI, project 7298. We acknowledge financial support from CNES (through the TOSCA project BANDINO) and the Embassy of France in Bangladesh. This work was also supported by the French

research agency (Agence Nationale de la Recherche; ANR) under the DELTA project (ANR-17-CE03-0001).





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
