# Peer review of "Storm surge hazard over Bengal delta: A probabilistic-deterministic modelling approach"

_Natural Hazards and Earth System Sciences, 2021_

## Referee Comment (RC1)

**Review of NHESS-2021-329**

**Title:** Storm surge hazard over Bengal delta: A probabilistic-deterministic modelling approach

The paper mainly has shown an application of the hydrodynamic model SCHISM combined with a large ensemble of synthetic storm events generated for the Bengal delta region based on the climatic conditions of the 1981-2015 period for flood risk analysis. I want to applaud the authors for their sheer volume of work. This study will undoubtedly contribute to the storm surge risk awareness of the coastal community and the region's local government; however, I have a few questions about some of the methods used in the analysis. Also, some sections throughout the paper need more details and better clarity to improve the model results' reliability before considering it for publication.

Questions are written below using page numbers and text quoted from the main body.

**Specific comments:**

1. The author has mentioned about a custom high-accuracy regional bathymetry data a couple of times. Is there any study done to check the quality of the data itself (i.e., accuracy)? Is it publicly available?

2. In page 12, lines 284-285: *'With an average annual frequency of 0.70314 cyclone, the ensemble of 3600 cyclones considered here ...'*
   What is the main reason behind selecting 3600 cyclones? Why this number?

3. These synthetic tracks are considered equivalent to 5000 years of storm activity to estimate the storm surge-induced water level at various return periods. Is it possible to identify any relationship between the return period of the maximum storm intensity and the surge water levels from the synthetic track properties? It could help recognize the role of different storm properties such as the maximum storm intensity and track translation speed in generating the surge level.

4. In page 9, lines 222-223: *'Following their findings, we have used a combination of Emanuel and Rotunno (2011) and Holland (1980) formulations to derive the parametric wind profile.'*

The Holland model's parametric wind and pressure field representation could generate unrealistic storm surge, especially when the track is slow-moving before the landfall and has higher intensity. As the study depends heavily on this atmospheric representation, it is crucial to show additional verification here. The author could add a comparison of the surge level generated during a historic event (e.g., cyclone Sidr), using a global reanalysis wind dataset (e.g., ERA5) and the parametric representation used in this study. This analysis can show the maximum surge bias using the parametric wind model and add more credibility to the later results.

5. In page 16, lines 367-369: *'The inundations in the mangrove region around 89.5°E seem to have a large saturation effect … slightly rising from 2.5m at 50-year return period to 3m at 500-year return period'*

If we compare the water level during Sidr in Figure 3 with Figure 6, we can see that close to 89.9 and 22 the floodwater during Sidr represents a 250-500 yr event. Is it realistic? How does it compare with the historical information?

6. In page 16, lines 382-383: *'Along the Hooghly estuary, the sensitivity of the water level to the return period is moderate for the first 100km but amplifies considerably further upstream.'*

Why do we see almost no change for 50 and 500 yr events at the downstream side, but then it shows a significant increase for the upstream part? How is this surge generating there?

7. In page 20, lines 429-431: *'We have first extracted the tidal water level from the 3600 cyclones that we simulated … our estimation of surge amounts to 1.8m at Hiron point.'*

This extraction needs to be explained better before going into the comparisons. How did the author separate the surge during high and low tides? Sometimes, the residual

(total water level - tides) during a low tide could be higher than the residual during a high tide, and it doesn't necessarily represent flooding.

8. In page 20, lines 441-442: *'However, the limited and potentially biased sampling of the "strongest" cyclones (17 in total, over 40 years) leads to an overestimation of the storm surge level.'*

   Jakobson et al. (2006) estimated the return period water levels using historic storms; here, we are looking at synthetic cases in this study. I don't think it is a fair comparison here.

9. In page 21, section 5.4: *'The maximum modelled water level reached about 5m around (88.4°E), which corresponds to a 250-year return period.'*

   Again, the statement here is from the results of this study. Can we verify it?

10. In page 22, section 5.6

    Before delving into this analysis, the author needed to show some model overland inundation comparisons with the high water mark data sets for a storm event. Otherwise, there is no way to verify this crucial information and could be misleading because of the potential inaccuracies in the topographic data.

   **Other comments**

1. In page 4, line 95: *'Additionally, the storm parameters ...'*

   What are these parameters?

2. In page 6, section 2.1

   The model runs are in 3D? If so, how many sigma layers are used? Also, how the Coriolis force is defined in the domain?

3. In page 10, line 248: *'... the time of landfall by half-an-hour to get a better match of the tidal propagation'*

   Surge propagation?

4. In page 11, Figure 3

   Please show the cyclone Sidr track on top of Figure 3a.

5. In page 13, lines 306-307: *'Second, with wind and pressure fields ... hindcasts described in the previous section.'*

   Do the tidal forcings also match the timeline of the synthetic storms?

6. In page 14, line 347: *'According to the polder embankments dataset used ...'*

   How are they incorporated into the current model setup?

7. In page 15, lines 356-357: *'... contrasting range of MHW along the shoreline – with two macrotidal poles ...'*

   A tidal MHW map needs to be added to Figure 5 as a subplot to illustrate this better. It will also help the description written in section 5.2.

8. In page 19, line 400: *'This landfall pattern corresponds to previous observations that the landfalling cyclones in the Bangladesh coastline tend to move north-eastward (Ali, 1996).'*

   Does the JTWC observed data also support this statement?

---

## Author Response (AR1)

**Response to the Comments by Reviewer # 1**

**Title:** Storm surge hazard over Bengal delta: A probabilistic-deterministic modelling approach

The paper mainly has shown an application of the hydrodynamic model SCHISM combined with a large ensemble of synthetic storm events generated for the Bengal delta region based on the climatic conditions of the 1981-2015 period for flood risk analysis. I want to applaud the authors for their sheer volume of work. This study will undoubtedly contribute to the storm surge risk awareness of the coastal community and the region's local government; however, I have a few questions about some of the methods used in the analysis. Also, some sections throughout the paper need more details and better clarity to improve the model results' reliability before considering it for publication.

Questions are written below using page numbers and text quoted from the main body.

**Specific comments:**

1. The author has mentioned about a custom high-accuracy regional bathymetry data a couple of times. Is there any study done to check the quality of the data itself (i.e., accuracy)? Is it publicly available?

Reply:

We would like to thank the reviewer for posing this very important question, as it is one of the main strengths of our modelling work.

**Dataset**

The initial version of the bathymetry was first developed by Krien et al. (2016) combining the following datasets -

1. 4 (four) navigational charts from National Hydrographic Office (NHO), and 60 (sixty) charts from Inland Waterways Authority of India (IWAI) for Hooghly river. They amount to 16,500 and 123,000 digitized sounding points respectively.
2. 1 (one) chart from the Bangladesh Navy, and 3 (three) charts from the Mongla Port Authority.
3. River cross-sections from Bangladesh water development board.
4. A 50-m resolution digital topography model developed by the Center for Environmental Geographic Information Services (CEGIS) through dedicated surveying.

This dataset is supplemented by GEBCO (2009) and ETOPO2 (from 89.3E to 92.3E and from 19.90N to 21N approximately).

When compared with the global GEBCO dataset, GEBCO is found to be much shallower compared to our dataset, on average by about 3m. The comparison is discussed in detail in Krien et al. (2016).

The dataset developed by Krien et al. (2016) was further updated by Khan et al. (2019) by adding 34 new Navigational charts collected from the Bangladesh Navy. This amounts to 77,000 new digitized points shown in Figure C1.

[Figure]

Figure C1. 77,000 digitized sounding points (in yellow). The coverage of the corresponding individual charts is shown in red outlines. The background image is taken from ESRI World Imagery services.

It is noteworthy here that, in Krien et al. (2016), embankments were also included as uniform height along the trace of the crests. For all embankments the same height was assumed (4.5m MSL). The embankment outlines were provided by the Bangladesh Water Development Board.

In the dataset assembled by Khan et al. (2019), which forms the basis of this paper, the crests heights still remain uniform over a single embankment but are replaced with respective measured (when available), or designed crest height for each embankment.

**Quality check**

As published in Krien et al. (2016), GEBCO global dataset is too shallow and has interpolation artifacts in the nearshore zones. GEBCO/STRM also shows very high topographic height over the mangroves (about 5m) due to the bias induced by the canopy (Shortridge and Messina 2011). As no other dataset is available from cross-validation, Krien et al. (2016) demonstrated the quality of the dataset by modelling the tide in the region. As tidal

propagation in the shallow water area is strongly controlled by the depth, a good regional reproduction of tide indirectly indicates the quality of the bathymetry.

The tidal model developed by Krien et al. (2016) shows 2-4 times improvement of the complex error compared to the state-of-the-art global tidal solutions, as well as consistent reproduction of tidal constituents (both amplitude and phase) all along the shoreline. The model once updated by Khan et al. (2019) shows error further reduced by 10-30% at most stations . These improvements are in large part due to the proper representation of the bathymetry, indirectly indicating its quality as mentioned above.

**Data availability**

The individual datasets used to derive this custom bathymetry can be queried from the original sources. We can not assume the right to redistribute the dataset freely, as we do not own the copyrights of the various sources of our composite product. Please note, however, that the oceanic part of the bathymetric data used in Krien et al 2016 is freely available.

2. In page 12, lines 284-285: 'With an average annual frequency of 0.70314 cyclone, the

ensemble of 3600 cyclones considered here …'

What is the main reason behind selecting 3600 cyclones? Why this number?

Reply:

The choice of the cyclone number is subjective, guided by the scale of the problem as well as the previous experiences of the authors. Indeed with the method of Emanuel et al. (2006), more (or less) number of cyclones could have been generated. That would have meant more computing requirements for both cyclone generation and storm surge computation.

As we have shown in Figure 4, this large number of cyclones allows a dense spatial sampling of cyclones, which in turn allows us to compute a large-range of the return period of storm surges with a high level of confidence.

3. These synthetic tracks are considered equivalent to 5000 years of storm activity to estimate the storm surge-induced water level at various return periods. Is it possible to identify any relationship between the return period of the maximum storm intensity and the surge water levels from the synthetic track properties? It could help recognize the role of different storm properties such as the maximum storm intensity and track translation speed in generating the surge level.

Reply: Given the large size of the sampling, our expectation is that it should shed some light on the role of various properties of the cyclone on the surge. For a given point of interest (POI), we have listed the following storm properties -

1. Storm intensity (wind and pressure)
2. Landfall location (left or right of the POI)

3. Approach angle with the coast
4. Translation speed of the storm

As discussed in the paper, as well as Krien et al. (2017), Khan et al. (2020) - the storm surge generated from a given storm characteristic will further be modulated by the following physical and hydrodynamical property -

1. Location of the POI (e.g., depth)
2. Phasing with tide, and tide-surge non-linear interaction

Because of this interaction and interdependence between tide, surge, total water level, and local bathymetry setting, understanding the role of various parameters is not straightforward to analyze or compare. Each analysis needs to be segmented into multiple bins of storm properties and multiple bins of water level for distangingling the tide. The analysis also would need to be performed segment-by-segment over the Bengal coastline.

Hence, the analysis regarding the role of the various storm parameters on the surge level would deserve a separate study by itself, which is beyond the scope of the present paper. We have plans to continue working on this large ensemble dataset to further disentangle the role of various factors and quantify their relative contribution and communicate our results in the future as a full-fledged article.

4. In page 9, lines 222-223: 'Following their findings, we have used a combination of Emanuel and Rotunno (2011) and Holland (1980) formulations to derive the parametric wind profile.'

The Holland model's parametric wind and pressure field representation could generate unrealistic storm surge, especially when the track is slow-moving before the landfall and has higher intensity. As the study depends heavily on this atmospheric representation, it is crucial to show additional verification here. The author could add a comparison of the surge level generated during a historic event (e.g., cyclone Sidr), using a global reanalysis wind dataset (e.g., ERA5) and the parametric representation used in this study. This analysis can show the maximum surge bias using the parametric wind model and add more credibility to the later results.

Reply:

We understand the concern of the reviewer, and upon reviewing the section in question, it is clear that all the details of our analysis were not presented. There are several folds of the answer to this query, and we will try to cover them thoroughly one by one.

**About the ability of reanalysis wind fields to capture the storms**

Our findings show that reanalysis fields over the Bay of Bengal are not well-resolved to get the extreme water level. To demonstrate that using reanalysis fields solely is not enough for hindcasts, we present a comparison of maximum wind speed and central pressure during

cyclone SIDR between JTWC Best Track (based on satellite observation), and two state-of-the-art reanalyses (ERA5 and CFSR v2) in Figure C2. This comparison clearly illustrates that both intensity parameters of the storms - wind and pressure - are reproduced poorly by the reanalyses, with a large underestimation of their intensity.

[Figure]

Figure C2. Comparison of Maximum wind speed (left) and Central pressure (right) during cyclone SIDR among JTWC Best Track (blue), ERA5 (orange), and CFSR (green).

This question was discussed in our previous paper (Khan et al. 2021) aimed at simulating the cyclone Amphan. We have tested storm surge modelling particularly for GFS, and we found that the GFS vastly underestimates the drop in the central pressure. The storm surge is not well reproduced for a GFS-only simulation. This is illustrated in Figure C5 and Figure C6 of the Author reply document of our article ( https://doi.org/10.5194/nhess-2020-340-AC1). The pdf file can be directly accessed here - https://nhess.copernicus.org/preprints/nhess-2020-340/nhess-2020-340-AR1.pdf

In other words, global reanalysis fields are too strongly biased to bear relevance for a meaningful assessment of an analytical wind and pressure field in a cyclone, hence no further revision is done in the manuscript in this regard.

**Bias in analytic wind field**

We agree with the Reviewer that there is some bias in individual analytical wind field formulas (Krien et al. 2018). The keyword here is 'individual'. We would like to clarify that we did not use solely the Holland model for wind, but an ad-hoc combination of Holland (1980) and Emanuel and Rotanno (2011) models based on radial distance, so as to minimize this bias. In the preprint, L302-304 reads as follows:

*"Here, for the inner core of the cyclones (r < r50 ), the wind field is estimated using the model of Emanuel and Rotunno (2011). On the outer core (r >= r50 ), we used the Holland (1980) model."*

This combination technique used here is based on the findings by Krien et al. (2018). By comparing the analytic fields with satellite scatterometer data, they showed that Emanuel and Rotanno (2011) model fits well to the inner core of the cyclone, and Holland (1980) fits well to the outer core. Hence, by combining these two analytical models the bias from a single analytical model (which is the concern raised in the review question) is reduced.

Although we discussed this combination technique in Section 2.3 and referred to Krien et al. (2018) there, the reference was left-out in the description of Section 3.2. We revised our manuscript to explicitly mention it again in Section 3.2 for clarity.

L302: "...as explained in Section 2.3. For the wind fields, we again employed a combination of parametric models based on the findings of Krien et al. (2018) to minimize the bias in individual parametric models. Here, for the inner core..."

**Simulation detail**

We have done two simulations for Sidr -

1. Same as Krien et al. (2017) with an analytic wind and pressure field in the inner core of the cyclone, and CFSR V2 dataset in the outer field of the cyclone. This combination of an analytic wind field and a background field from an atmospheric model gave a good reproduction of the wave fields at two buoys in the Bay of Bengal (see Krien et al. 2017). Similar results were found by Khan et al. (2021) for cyclone Amphan - where the analytical field was combined with GFS fields.
2. A sensitivity simulation of Sidr, only with an analytical field, to illustrate that the peak water level is captured.

Indeed, we can follow the strategy of combining an analytical field with a modelled atmospheric field in case of an observed/ongoing cyclone (Krien et al. 2017, Khan et al. 2021). But for the ensemble, no such atmospheric model field can be accessed - hence can not be merged. The objective of the Sidr simulation with only the analytical wind field is to illustrate that the analytical wind field captures the water level peak correctly around the landfall location. Hence, we purposefully used a simulation of Sidr with only an analytical wind field (Simulation 2 above) for consistency with the ensemble simulations.

We revised the corresponding text (L243-251) to clarify the simulation details of cyclone Sidr, as follows -

"To supplement the results shown in Khan et al. (2021a), we also performed a hindcast of the water level generated by cyclone Sidr – another comparatively well-documented historic cyclone (Krien et al., 2017b). Figure 3 shows the water level evolution at four tide gauge stations around the landfall location. The modelled water level is shown in black lines, and

the observed water level records are shown in dots. As noted by Krien et al. (2017b), the tide gauge stopped working at Khepupara at the time of very high water. The error of the peak modelled water level typically amounts to 10-20 cm. It is to be noted that, in their hindcast, Krien et al. (2017b) shifted the time of landfall by half-an-hour to get a better match of the tidal propagation. Here we retained the original JTWC track hence the slight shift in phase of the water level. Moreover, both Krien et al. (2017b) and Khan et al. (2021a) combined their analytical wind and pressure field (around the storm center) with a background field (far from the storm center) taken from global atmospheric models. Here we used only the analytical wind fields to be consistent with the forcing strategy used for the storm surge computation from the cyclone ensemble (Section 3). The results from this experiment illustrated in Figure 3 show that the peak water levels around the landfall are well captured. This peak water level is the main variable we will deal with in our storm surge hazard assessment."

5. In page 16, lines 367-369: 'The inundations in the mangrove region around 89.5°E seem to have a large saturation effect ... slightly rising from 2.5m at 50-year return period to 3m at 500-year return period'

If we compare the water level during Sidr in Figure 3 with Figure 6, we can see that close to 89.9 and 22 the floodwater during Sidr represents a 250-500 yr event. Is it realistic? How does it compare with the historical information?

Reply:

There is unfortunately no observational historical record to check whether our estimate is correct. However, the consistent behaviour of our model and data throughout the Bengal coastline indicates the estimates are reasonable. Our reasoning is as follows -

First, Sidr is one of the strongest events in recent decades, with an estimated death toll of 15000 people. Sidr was an Extremely Severe Cyclonic Storm (Indian Meteorological Department classification). With peak 1-minute sustained winds of 260 km/h (about 140 knots), it was a Category-5 equivalent tropical cyclone on the Saffir–Simpson scale. If we consider the distribution of Figure 4(b) in the manuscript, Sidr was a very rare event in the record.

Second, the Bengal delta coastline is long, and the flooding pattern is primarily restricted to the surroundings of the landfall location (Khan et al. 2021). There are other historical events - such as the 1970 Bhola cyclone which took 500000 lives, or the 1991 Bangladesh cyclone (also known as Gorky) which took 138000 lives. These cyclones made landfall over the Chittagong coast (over 200km to the east of the Sundarbans mangroves) - where the tidal range is high and our results also show significantly higher total water levels at all return periods. In other words, our method is predicting high water levels, where historically multiple very large storm surges were recorded.

Third, our modelling platform has been tested previously in Krien et al. (2017) and Khan et al. (2021) and performed well in this region. We also show in the manuscript that the storm

climatology in our ensemble is consistent with the available observed climatology (Figure 4). Hence, we tend to trust the estimate of return level we get from these models and data. Additionally, mangroves are known to damp the tidal-surge waves by acting as an energy damping zone (implemented as manning friction in our model). Hence, the saturation we have seen is expected and our modelling result is showing as such.

In a brief, using a realistic storm surge model forced with the realistic distribution of storms, we can expect the output to be realistic too. The objective of this paper is to fill in the knowledge gap of the storm surge hazard in the study region that arises from the absence of historical data, as stated in the Introduction section (see L59-65).

6. In page 16, lines 382-383: 'Along the Hooghly estuary, the sensitivity of the water level to the return period is moderate for the first 100km but amplifies considerably further upstream.'

Why do we see almost no change for 50 and 500 yr events at the downstream side, but then it shows a significant increase for the upstream part? How is this surge generating there?

Reply:

First of all we would like to point out that the keyword here is water level, not surge. As we have explained in our manuscript (L35-40, L121-127), due to non-linear tide-surge interaction, the water level is not a linear combination of tide + surge + wave setup but rather a non-linear, dynamic one. We have thus consistently discussed the water level, instead of surge (surge being defined as total water level minus tidal prediction L20). In Figure 7, we have shown the amplification of the water level estimate with respect to the water level at the 50-year return period.

The reason behind this is not entirely clear to us, but one possible reason is the amplification of higher levels of surges. In our 2020 paper (Khan et al. 2020), we show that tidal properties change substantially along the Hooghly estuary. To add further justification to our reasoning, we analyse two points from Figure 7 (original manuscript) - Sagar Roads (located at 25km) and Diamond Harbour (located at about 100km).

[Figure]

Figure C3. (a) Total water level at the given return periods for Sagar Roads (blue) and Diamond Harbour (red). The solid lines are from the ensemble simulation with coupled tide, surge and wave and the dashed lines are only coupled tide and surge but without waves. (b) Only surge level as multiple of 50-year surge level for Sagar Roads (blue) and Diamond Harbour (red). The surge level is simulated through the coupled wave model but without forcing any tide, i.e., always at mean sea level (MSL).

First, we analyse the total water level at various return periods for these two locations (Figure C3a). Two modelling configurations are used - one with full coupling of tide-surge-waves (solid line) and another is only tide-surge but without waves (dashed line). From the result, the contribution of waves in total water level is clear. The contribution varies among the two stations, but in general the amplification along the estuary remains the same as what we have reported in our manuscript - e.g., total water level amplifies after 100km compared to 50-year return period water level. This in turn indicates that the wave setup is not the component causing this amplification. Then the component that remains is the tide.

Second, to see if the tide is creating the upstream amplification, we extracted the surge estimate from the tide-free version of the storm simulations ensemble. The mean water level is fixed at 0m MSL for these simulations. The surge at various return periods is shown in Figure C3b, as a multiplication of the 50-year surge level. We see that between upstream and downstream the evolution of surge with a return period remains practically the same. This indicates the evolution of the total water level shown in Figure 7 is caused by the non-linear combination of tide, surge and wave-setup.

We have also revised our manuscript:

"Along the Hooghly estuary, the sensitivity of the water level to the return period is moderate for the first 100km but amplifies considerably further upstream. This amplification

appears to be linked to the changes in the tide along the estuary (See Supplementary materials)."

7. In page 20, lines 429-431: 'We have first extracted the tidal water level from the 3600 cyclones that we simulated ... our estimation of surge amounts to 1.8m at Hiron point.'

This extraction needs to be explained better before going into the comparisons. How did the author separate the surge during high and low tides? Sometimes, the residual (total water level - tides) during a low tide could be higher than the residual during a high tide, and it doesn't necessarily represent flooding.

Reply:

We believe that by the separation of tide and surge the reviewer is referring to the issue of tide-surge interaction and their dependence. In practice, the surge is computed classically as the difference between the total water level and pure tidal water level. This mathematical definition means that, often a high surge occurs during a low-tide, but does not actually cause flooding. To alleviate such issues, new metrics such as the skew-surge have been proposed (e.g. de Vries et al., 1995), which computes surge as a difference between maximum water level and nearest maximum tidal water level.

However, the section indicated in the comment focuses on comparing with a published result by Lee (2013), as such it is necessary to analyse similar variables. As we have explained in the manuscript - L422-433, Lee (2013) used Ensemble Empirical Mode Decomposition to remove the tidal signal and for detrending the water level. Once the tidal signal is removed, they use the yearly block maxima to perform an Extreme Value Analysis by fitting into the GEV distribution. To have the maximum compatibility with the random variable used by Lee (2013) (e.g. maximum non-tidal residual), we have done the following as explained in the manuscript (L428) -

1. Computation of tidal water level for the full ensemble of cyclones.
2. Computation of the temporal maximum of total water level minus tidal water level to get a set of maximum surge values for the storm ensemble. This is the closest to what Lee (2013) analysed, in terms of the random variables.
3. Computation of the statistics as described in Section 3.3

In light of the additional confidence interval estimate, we have revised the corresponding sections in the manuscript. The sections now read as follows -

"Using a yearly-maximum method, in his extreme value analysis, he obtained a 1.66m [1.50-1.95] surge level at 50-year return period, and 1.75m [1.57-2.14] at 100-year return period. The range in the parenthesis is the 90% confidence interval.

In the previous section, our analysis was focused on the water level rather than surge level. To compare with the estimate of Lee (2013), we reprocessed the whole ensemble of storm event simulation results. We have first extracted the tidal water level from the 3600 cyclones that we simulated. Then, for each cyclone, we extracted the maximum surge level. Finally, on this maximum surge estimate, we applied the same ranking-based return period estimate. At Hiron point, our estimation of surge amounts to 1.77m [1.68-1.85] at 50-year, and 2.31m [2.12 - 2.47] at 100-year return period. The range in the parenthesis is the 90% confidence interval. At 50-year return period, with a difference of only 14cm (inferior in Lee (2013)), our estimated value is comparable to the estimated value by Lee (2013) from the observation time series. The confidence interval was about twice narrower compared to Lee (2013). To be noted that, the estimated 50-year return period water level from our ensemble is about 3m at Hiron point. At 100-year return period the estimate of Lee (2013) was underestimated compared to ours by 56cm. See supplementary for further details."

8. In page 20, lines 441-442: 'However, the limited and potentially biased sampling of the "strongest" cyclones (17 in total, over 40 years) leads to an overestimation of the storm surge level.'

Jakobson et al. (2006) estimated the return period water levels using historic storms; here, we are looking at synthetic cases in this study. I don't think it is a fair comparison here.

Reply: We agree this statement lacked clarity. The objective of our discussion was to compare with all the available previous results. Indeed, Jakobsen et al. (2006) tried to establish a return period of storm surge based on modelling along the Bengal coastline, and our objective was in no way to criticise this valuable work.

Saying that, Jakobsen et al. (2006) themselves mentioned that they analysed "17 severe cyclones", and identified it as a limitation of their study. We think that our comparison is fair as we have voiced the same limitation as the authors themselves in their original paper. Additionally, it is needed in the manuscript for comprehensiveness.

We propose to update the word "strongest" with "severe" taking verbatim the wording from Jakobsen et al. 2006 (L441).

9. In page 21, section 5.4: 'The maximum modelled water level reached about 5m around (88.4°E), which corresponds to a 250-year return period.'

Again, the statement here is from the results of this study. Can we verify it?

Reply:

We confirm the result is from our study. The results are shown in Figure 3 for Sidr, and Figure 6(g) for the 250-year return period water level. We humbly point out that no such validation

dataset exists for return period estimates of water level, and we hope to fill in this knowledge gap through this paper itself (Please see Introduction section, L59-65).

We revise our manuscript as following -

"The maximum modelled water level reached about 5m around (88.4°E) (Figure 3a), which corresponds to a 250-year return period (Figure 7g)."

Please note that Figure 7g refers to Figure 6g in the initial manuscript.

10. In page 22, section 5.6

Before delving into this analysis, the author needed to show some model overland inundation comparisons with the high-water mark data sets for a storm event. Otherwise, there is no way to verify this crucial information and could be misleading because of the potential inaccuracies in the topographic data.

Reply:

We agree the potential inaccuracies of the topography is a very relevant issue. Unfortunately again, no such consolidated dataset of high water mark exists for comparison, to the best of our knowledge. Whenever we could find some information (e.g. Islam et al. 2011 ), it was highly unreliable due to missing datum references.

We have discussed this just before we delve into the inundation exposure (Section 5.5). We also indicate the limitation of our estimate regarding the possible existence of city protection embankments (beside the coastal protection embankments included in our model), for which data is not publicly available and not included in our model (L526-528). We also give a reminder about this limitation in the conclusion (L555-559).

**Other comments**

1. In page 4, line 95: 'Additionally, the storm parameters ...'

What are these parameters?

Reply: We propose to update the line as following -

Additionally, the storm parameters - notably maximum wind speed, radius of maximum wind, and central pressure - are collected over a long period, and the homogeneity of the storm records is not well defined (Singh et al. 2020).

2. In page 6, section 2.1

The model runs are in 3D? If so, how many sigma layers are used? Also, how the Coriolis force is defined in the domain?

Reply:

We are sorry for not having mentioned this information explicitly in our manuscript.

The model configuration used in this study is a 2D barotropic one. Past studies demonstrated the model's capability to predict storm surge quite accurately in 2D barotropic mode (Krien et al. 2017, Khan et al. 2021).

As the model is discretized in spherical coordinates, the coriolis parameter is variable over the domain, computed locally at nodes, based on latitude using the exact expression $f = 2\omega \sin(\varphi)$.

We propose the following revision for the model description.

L161: "...built from the original SELFE (Zhang and Baptista, 2008). SCHISM solves the standard Navier-Stokes equations with hydrostatic and Boussinesq approximations in an unstructured grid, which can be discretized using a triangular or hybrid triangular-quad element. SCHISM also includes..."

L174: "... its surroundings (FIgure 2a). The model mesh is defined in latitude-longitude (spherical grid). We have used variables..." "... 1.1 million triangular elements." "..." "...The model transforms the coordinates and most of the calculation is done on a local frame. For all simulations described in this article, we have used a 2D barotropic configuration, which is shown to well-reproduce the tide and storm surges (Bertin et al. 2014, Krien et al. 2017, Khan et al. 2021). A time step of 300 seconds..."

3. In page 10, line 248: '... the time of landfall by half-an-hour to get a better match of the tidal propagation.'

Surge propagation?

Reply:

We agree our statement was misleading. We meant tide-surge evolution - e.g., total water level. Krien et al. (2017) postulates that the landfall timing reported by JTWC was off by half an hour, which in turn stimulated their model as a shifted peak water level.

We propose to replace "better match of the tidal propagation" with "a better match of the timing of the peak water level."

4. In page 11, Figure 3

Please show the cyclone Sidr track on top of Figure 3a.

Reply:

The track is now shown. The figure is updated as shown in Figure C4.

[Figure]

Figure C4. Updated Figure 3.

5. In page 13, lines 306-307: 'Second, with wind and pressure fields ... hindcasts described in the previous section.'

Do the tidal forces also match the timeline of the synthetic storms?

Reply:

Yes, the tidal forcing matches the timeline of the synthetic storms. However, the synthetic storms only have day, month, and hour (being a climatology of storms by design), but do not have any year attached to it. We have randomly chosen the year for each cyclone, with equal probability for each year in the climatology period 1980-2015.

We have revised our manuscript as follows -

"... and Kharnaphuli (Chowdhury and Al Rahim, 2012). Applied tidal forcing and the tidal water level boundary are also consistent with the timestamps of the synthetic cyclones. Similar to the model setup ..."

6. In page 14, line 347: 'According to the polder embankments dataset used ...'

How are they incorporated into the current model setup?

Reply:

Reviewer 2 also posed a similar question. Our regular grid bathy-topo dataset (with highest resolution of 50m) does not capture these embankments. The width of the embankment in reality is in the order of 10-20m. The original embankment datasets are provided by the Bangladesh Water Development Board (BWDB) as line shapefiles. The correct characterization of the embanked areas would be - polder - e.g, a region surrounded by embankments. One such region is shown in Figure C5. In local use, polder became synonymous with the embanked region as well as the embankments themselves.

To incorporate the shapefile dataset into the grid, from the original line shapefiles, a buffered polygon is first created. The lines from the shapefile are taken as the outer edge of the polder during this process.  The buffer size is controlled by the target mesh resolution. In our case, we took a buffered area of 300m - and assigned all the mesh node points inside this buffered area to have the height corresponding to the respective embankment height. Additionally, during mesh generation, we have forced the mesh generator to follow the embankment shapefile while assigning position of the nodes. This process assigns the embankment heights along the edge of an element. This process sets the embankment height to one or two nodes of a triangular element, sometimes all three (for example, when the embankment has a strong curve, or two side-by-side embankments are separated by sub-resolution distance.).

In this study we used 10cm as the threshold for the wetting-drying algorithm. That means, if all nodes of a given mesh element have a water level above 10cm, the element is considered wet, otherwise dry.

[Figure]

Figure C5. Embankments as incorporated into the model. Snapshot for Haitya Island (91.11°E, 22.24°N).

We have also revised the manuscript with the following lines -

"...1.1 million triangular elements. In order to take into account the embankments information, we have aligned the mesh nodes along the contour of the embankments and set the height values of these nodes to the dikes levels provided by BWDB. The flow above the top of the embankments is controlled by the wetting-drying algorithm of SCHISM just like everywhere else over the modelling domain. The model transforms..."

7. In page 15, lines 356-357: '... contrasting range of MHW along the shoreline – with

two macrotidal poles ...'

A tidal MHW map needs to be added to Figure 5 as a subplot to illustrate this better. It will also help the description written in section 5.2.

Reply:

We thank the reviewer for this suggestion. We have updated Figure 5 with a tidal Mean High Water map.

[Figure]

Figure C6. (a) Inundation extent and corresponding water level at 50-year return period. Black star shows the tide gauge location where the confidence interval is reported in Figure 6. (b) Mean High Water (MHW) derived from a year-long tidal simulation. (c) Water level for the 50-500 years return period expressed as a multiple of the MHW level along the nearshore dash-dotted line shown in (a) and (b).

We have also updated corresponding lines in the manuscript. In particular, now L337 reads as "The 50-year return period water level shows a similar spatial pattern as the mean high water, as shown in Figure 5b, as well as the tidal range (Khan et al. 2021)"

8. In page 19, line 400: 'This landfall pattern corresponds to previous observations that the landfalling cyclones in the Bangladesh coastline tend to move north-eastward (Ali, 1996).'

Does the JTWC observed data also support this statement?

Reply:

This statement is supported by the JTWC dataset and discussed in further detail supplemented with numerical modelling in a recent article by Akter and Tsuboki (2021), for the storms recorded during the 1990-2019 period. The same conclusion was also drawn by Mondal et al. (2021). We updated our manuscript to add these references as following -

"This landfall pattern corresponds to previous observations that the landfalling cyclones in the Bangladesh coastline tend to move north-eastward (Ali 1996, Akter and Tsuboki 2021, Mondal et al. 2021)."

**References**

Akter, N. and Tsuboki, K., 2021. Recurvature and movement processes of tropical cyclones over the Bay of Bengal. Quarterly Journal of the Royal Meteorological Society, 147(740), pp.3681-3702.

Bertin, X., Li, K., Roland, A., Zhang, Y.J., Breilh, J.F. and Chaumillon, E., 2014. A modeling-based analysis of the flooding associated with Xynthia, central Bay of Biscay. Coastal Engineering, 94, pp.80-89.

de Vries, H., M. Breton, T. de Mulder, Y. Krestenitis, J. Ozer, R. Proctor, K. Ruddick, J. Salomon, and A. Voorrips, 1995. A comparison of 2D storm surge models applied to three shallow European seas, Environ. Software, 10(1), 23–42, doi:10.1016/0266-9838(95)00003-4.

Islam, A.S., Bala, S.K., Hussain, M.A., Hossain, M.A. and Rahman, M.M., 2011. Performance of coastal structures during Cyclone Sidr. Natural Hazards Review, 12(3), pp.111-116.

Mondal, M., Biswas, A., Haldar, S., Mandal, S., Bhattacharya, S. and Paul, S., 2021. Spatio-temporal Behaviours of Tropical Cyclones over the Bay of Bengal Basin in last Five Decades. Tropical Cyclone Research and Review, doi: 10.1016/j.tcrr.2021.11.004

**Response to the Comments by Reviewer # 2**

Comment on nhess-2021-329

Anonymous Referee #2

Referee comment on "Storm surge hazard over Bengal delta: A probabilistic-deterministic modelling approach" by Md Jamal Uddin Khan et al., Nat. Hazards Earth Syst. Sci. Discuss., https://doi.org/10.5194/nhess-2021-329-RC2, 2021

I commend the authors on this mammoth work on a topic that is highly relevant and much needed for Bangladesh and this region. The work has been performed well and described thoroughly. I believe that the paper will be ready to publish after the authors clarify a few doubts and questions.

The issues that require clarification center on two points: 1) the embankments ; 2) the cyclone ensemble and return periods.

Re. the embankments:

1. The authors should explain exactly how the information on embankments has been integrated and should provide a map and/or more information showing embankment heights within their dataset (this can be smoothed out if there are copyright or other data concerns). As the authors state, these embankments are critical in controlling flooding.

However, I do not know if these embankments are wider than 250 m, and it is not clear if/how these can be captured by the bathy-topo datasets if the minimum resolution is 250 m.

Reply:

The bathy-topo dataset used in this study is an assembly of several bathy-topo dataset for which the maximum resolution amounts to 50m and indeed does not capture these embankments as their typical width is 10-20m. The original embankment datasets were provided by Bangladesh Water Development Board (BWDB) as polygon shapefiles. A figure showing the embankment outlines and heights are now added in the supplementary materials. The correct characterization of the embanked areas would be - polder - e.g, a region surrounded by embankments. One such region is shown in Figure C1. In local use, the polder became synonymous with the embanked region as well as the embankment themselves.

To incorporate the shapefile dataset into the grid, from the original line shapefiles, a buffered polygon is first created. The lines from the shapefile are taken as the outer edge of the polder during this process. The buffer size is controlled by the target mesh resolution. In our case, we took a buffered area of 300m - and assigned all the mesh node points inside this buffered area to have the height corresponding to the respective embankment height. Additionally, during mesh generation, we have forced the mesh generator to follow the embankment shapefile while assigning the position of the nodes. This process assigns the

embankment heights along the edge of an element. This process sets the embankment height to one or two nodes of a triangular element, sometimes all three (for example, when the embankment has a strong curve, or two side-by-side embankments are separated by sub-resolution distance.).

In this study we used 10cm as the threshold for the wetting drying algorithm. That means, if all nodes of a given mesh element have a water level above 10cm, the element is considered wet, otherwise dry

[Figure]

Figure C1. Embankments as incorporated into the model. Snapshot for Haitya Island (91.11°E, 22.24°N). Here the colour scale represents the bathymetry, and the red pixels are above 4mMSL.

We have also revised the manuscript with the following lines -

"…1.1 million triangular elements. In order to take into account the embankments information, we have aligned the mesh nodes along the contour of the embankments and set the height values of these nodes to the designed dikes levels provided by BWDB. The flow above the top of the embankments is controlled by the wetting-drying algorithm of SCHISM just like everywhere else over the modelling domain. The model transforms…"

2. The authors mention that the embankments appear to start overflowing at the 75-100 year RP water levels. Clarification on how the model "sees" these embankments will be

useful. I assume that overtopping processes are not included in the model but this will be worth stating.

Reply:

In application, the embankments are seen realistically as a series of sharp-crested blocks. The flow over the embankments is controlled by a wetting and drying scheme, just like everywhere else in the model domain. The water can pass freely from one element (outside an embankment) to another element (inside an embankment) when all of the nodes in the first element are registered as "wet".

The model does account for the surface waves through its online coupling between SCHISM and WWMIII. Hence, we do account for the overflowing driven by wave setup. However, the waves being modelled in a phase-averaged framework (spectral), the model does not account for the swash and associated overtopping. To take into account your comments about the embankment we have added the following sentence to clarify the way the embankments have been taken into account.

"… 1.1 million triangular elements. In order to take into account the embankments information, we have aligned the mesh nodes along the contour of the embankments and set the height values of these nodes to the dikes levels provided by BWDB. The flow above the top of the embankments is controlled by the wetting-drying algorithm of SCHISM just like everywhere else over the modelling domain. The model transforms the coordinates…"

Re. the cyclone ensemble:

3. I understand and support the authors' decision to show water level variations based on RPs rather than events. While logical, this can however be confusing to interpret. The authors should add a few sentences explaining describing how the 100 year RP water level map (for example) is comprised of WLs from several cyclone events.

Reply:

We agree. We propose to add the following revision in the description of our 50-year return period map to make it clearer.

L337: "…topography. This water level at the 50-year return period is computed from the full ensemble of cyclone simulations using the empirical statistical method described in Section 3.3, pixel-by-pixel. As such, the total water level at the 50-year return period has contributions from hundreds of different cyclones of our ensemble. In our estimate, about 600 individual cyclones contribute to the 50-year return period water level over the illustrated region. ¶ The 50-year return period water level shows a …"

Note: ¶ means a new paragraph.

Reply:

Note: Based on the referred figure, we interpret that you meant the opposite, as the spatial density of cyclones landfalls is superior in the east than in the west of the delta.

**Cyclone distribution**

We agree that the western segment (India coastline) has a lower number of cyclones per 20km box compared to the eastern segment (south-eastern Bangladesh). However, the density remains high in the west, in the range of 50-80 cyclones per 20km box along the shoreline. This results in a robust estimation of the water levels at a given return period, as shown by the confidence intervals that we incorporated in the revised manuscript (Figure 6a, d). The confidence interval is computed using a bootstrap method by sampling from the full ensemble with replacement. Hence, it also comprises the answer to the question regarding how the spatial variation of the cyclone tracks can influence the spread and uncertainty around the WL results at the shoreline and inland.

The number of cyclones south of 20°N does get significantly lower than the north of it. It is because of the selection process of the cyclone during the generation of the ensemble, where only the cyclones passing through a 300km radius circle centered on the central Bengal coast are retained. This selection criterion is due to the fact that our study has a focus on the Bengal delta only. We have already acknowledged this selection (L284), and in the results section, we limited all our figures analysing the storm surges to the north of 20°N.

**Water level range**

The water level range is not solely controlled by the storm surge but through a non-linear combination of tide, surge, and wave setup, which are all strongly controlled by local topography/bathymetry. We have stressed this point in our manuscript (L121-126), and we referred to Khan et al. 2021, and Krien et al. 2017 as well.

The hydrodynamic configuration is not the same for the East vs West side, as evident from the mean tidal range. The mean tidal range in the western side - around 87.75E, 21.5N - is slightly smaller (3m) than on the western side (4m) - around 91.75E, 22.25N (Khan et al. 2020). The same is seen for mean high water, shown in the revised Figure 5b. As such, it is not straightforward to comment on what part of the water level extremes results from the

local hydrodynamics and what part results from the difference in storm distribution. One can also note that in Figure 4, the central part of the delta around 89.5E shows a similar density of cyclone landfalls as the eastern side, however, the total water level at the 50 year return period (Figure 5) is hardly half as much. This goes to show that a smaller range of water level can not be unequivocally explained by a lower density of cyclones' landfall.

**References**

Khan, Md Jamal Uddin, et al. "Sea level rise inducing tidal modulation along the coasts of Bengal delta." Continental Shelf Research 211 (2020): 104289.

Khan, M. J. U., Durand, F., Bertin, X., Testut, L., Krien, Y., Islam, A. K. M. S., Pezerat, M., and Hossain, S.: Towards an efficient storm surge and inundation forecasting system over the Bengal delta: chasing the Supercyclone Amphan, Natural Hazards and Earth System Sciences, 21, 2523–2541, https://doi.org/10.5194/nhess-21-2523-2021, 2021.

Krien, Y., Testut, L., Islam, A., Bertin, X., Durand, F., Mayet, C., Tazkia, A., Becker, M., Calmant, S., Papa, F., Ballu, V., Shum, C., and Khan, Z.: Towards improved storm surge models in the northern Bay of Bengal, Continental Shelf Research, 135, 58–73, https://doi.org/10.1016/j.csr.2017.01.014, 2017.

**Response to the Comments by Reviewer # 3**

Comment on nhess-2021-329

Anonymous Referee #3

Referee comment on "Storm surge hazard over Bengal delta: A probabilistic-deterministic modelling approach" by Md Jamal Uddin Khan et al., Nat. Hazards Earth Syst. Sci.

Discuss., https://doi.org/10.5194/nhess-2021-329-RC3, 2021

The manuscript by Khan and co-authors addresses the problem of storm surge hazard assessment over Bengal delta. They propose a hybrid procedure that combines high resolution numerical simulations, ensemble modelling, and probabilistic analysis. Their results are of high interest for coastal risk planning in this region both for present-day and future climate conditions.

**Main comment**

The manuscript is well organized and the presentation of the methods and results are very clear. The conclusions are sound. I very much appreciate the efforts made by the authors to discuss their results with respect to existing studies and to the limitations of their work (Sect. 5). My background is mainly focused on statistics. Therefore I won't comment on the modelling part (Sect. 2). Regarding the statistical aspects, a few aspects should be clarified and further elaborated before publication. Therefore, I recommend additional corrections by incorporating, if possible, the following recommendations.

Reply: We thank the Reviewer for his/her commendation and useful advice to improve the statistical analysis presented in the manuscript.

**1) Return level estimates**

1.1. The authors stress several times in the manuscript that the estimate of the return period of the water level is 'robust'. I agree with them that with a dataset representing more than 5000 years of cyclone activity, robustness should be achieved for estimating 100-year return levels. However, for a 500-year return level, some statistical uncertainty could still affect the result. This should be analyzed more carefully. As far as I understand the procedure, the authors calculate the empirical percentile using the results of the ensemble ('ranking-based statistical analysis' as indicated in line 321). I would expect the authors to calculate some confidence intervals, for example using bootstrap approaches; in particular, the results in Figure 8 should be further discussed in relation to this additional uncertainty estimate. An additional interest is to support the discussion in Sect. 5.3, in particular for the comparison with the study of Leijnse et al. who provides such uncertainty estimates.

Reply:

**Confidence interval computation**

We agree with the Reviewer regarding our unbacked claim about robustness. Following the suggestion, we have computed the 95% confidence interval for the return level estimate using bootstrap method. The bootstrap computation is applied over the full domain, node-by-node. For each node, 10000 instances of sampling were drawn to compute the bootstrap and derive the confidence interval.

We have revised the Statistical analysis section as following -

"…5120 years (for the largest return period). We have used bootstrap technique to compute the confidence interval for the return level estimates (Hesterberg, 2011). Unless otherwise stated, we have pooled 10000 bootstrap samples of the same size as our ensemble (3600 cyclones) with replacements and applied the above mentioned ranking method without considering ties. ¶Estimated quantities at return periods…"

(Note: ¶ means a new paragraph.)

We choose to show the robustness of the storm surge estimate at a few relevant stations. A new figure is now added, Figure 6 in the revised manuscript, as shown in Figure C1 below. The confidence interval remains moderate for the features that have been discussed throughout the manuscript, and none of our conclusions in the original manuscript were altered.

[Figure]

Figure C1. Extreme water level evolution with return period at (a) Sagar Roads, (b) Hiron Point, (c) Charchanga, (d) Chittagong, (e) Diamond Harbour, (f) Chandpur. These station locations are shown in Figure 5. The shaded grey area indicates the 95% confidence interval.

The addition of Figure C1 is reflected in the manuscript as following -

"…above topography. This water level at the 50-year return period is computed from the full ensemble of cyclone simulations using the empirical statistical method described in Section 3.3, pixel-by-pixel. As such, the total water level at the 50-year return period has contributions from hundreds of different cyclones of our ensemble. In our estimate, about 600 individual cyclones contribute to the 50-year return period water level over the illustrated region. To supplement subsequent discussions, in Figure 6 we also present the water level estimates and corresponding 95% confidence intervals at few station locations for a range of return periods (upto 500 years). These station locations are shown in black stars in Figure 5."

"…delta (barely 3m). At 50-year return level, these estimates of water levels are objectively robust along the open coastlines, as well as inside the estuaries, with a couple of cm range in the computed 95% confidence interval (Figure 6)."

The station locations in Figure C1 are indicated in Figure 5. Revised Figure 5 is as shown in Figure C2 below -

[Figure]

Figure C2. (a) Inundation extent and corresponding water level at 50-year return period. Black star shows the tide gauge location where the confidence interval is reported in Figure 6. (b) Mean High Water (MHW) derived from a year-long tidal simulation. (c) Water level for the 50-500 years return period expressed as a multiple of the MHW level along

the nearshore dash-dotted line shown in (a) and (b).

**Update to Population Exposure Computation**

We have also updated Figure 8 (current Figure 9) with the confidence interval. The updated figure is shown in the reply to the next comment (Comment 2: Population exposure).

**Update to Comparison with Previous studies in the Discussion**

We have updated the text regarding the comparison with Leijnse et al. The current text reads as following -

"Leijnse et al. (2021) used a somewhat similar approach to ours. They used an extreme value analysis on the surge estimate of 1000 year simulated cyclonic activity using Peak-over-Threshold method and an exponential fit. In their estimate, the surge level (from tide free simulations) at Charchanga and Chittagong at 100-year return period is about 2.8m [2.5-3.1] and 3.3m [3.1-3.6] , respectively. The range in the parenthesis is the 95% confidence interval. To better compare with the estimate of Leijnse et al. (2021) we have re-simulated the ensemble of our 5000 year cyclonic activity (3600 cyclones) again, but without incorporating the tide. In our estimate, the 100-year return period surge level is 3.6m [3.4-4.0] and 4.1m [3.8-4.3] for Charchanga and Chittagong respectively. In other words, in these two locations, the estimate of Leijnse et al. (2021) is more than 60cm lower than ours, while the confidence interval range between there and this study are essentially the same."

We have also updated the comparison with Lee (2013), who provided a 90% confidence interval. The corresponding texts now reads as follows -

"Using a yearly-maximum method, in his extreme value analysis, he obtained a 1.66m [1.50-1.95] surge level at 50-year return period, and 1.75m [1.57-2.14] at 100-year return period. The range in the parenthesis is the 90% confidence interval.

In the previous section, our analysis was focused on the water level rather than surge level. To compare with the estimate of Lee (2013), we reprocessed the whole ensemble of storm event simulation results. We have first extracted the tidal water level from the 3600 cyclones that we simulated. Then, for each cyclone, we extracted the maximum surge level. Finally, on this maximum surge estimate, we applied the same ranking based return period estimate. At Hiron point, our estimation of surge amounts to 1.77m [1.68-1.85] at 50-year, and 2.31m [2.12-2.47] at 100-year return period. The range in the parenthesis is the 90% confidence interval computed from 10000 bootstrap samples. At a 50-year return period, with a difference of only 14cm (inferior in Lee (2013)), our estimated value is comparable to the estimated value by Lee (2013) from the observation time series. The confidence interval was

about 2 times tighter compared to Lee (2013). To be noted that the estimated 50-year return period water level from our ensemble is about 3m at Hiron point. At the 100-year return period, the estimate of Lee (2013) was underestimated compared to ours by 56cm. See supplementary for further details."

1.2. A second aspect is the comparison of return levels to observed surge levels during cyclones. Could the authors consider the relevancy of using the surgedat dataset to this aim? http://surge.climate.lsu.edu/data.html

Reply:

We have checked out the dataset, and it was interesting. Figure C3a shows the data points that were reported to the same map extent as Figure 4 of the manuscript.

[Figure]

Figure C3. (a) Spatial distribution of the Storm tide in surgedat dataset. Colorbar represents the Storm tide value in the dataset. (b) Time series of the Storm tide. Missing values are not shown (e.g., 2007). (c) Year distribution of the record in surgedat dataset.

From the initial check, it seems that some values are suspiciously high. These values are mostly recorded during the 1950s-1970s. Through personal communication, we got confirmation from the curator of the dataset, Dr. Barry Keim, that the datums of these values are not uniform, often unknown. It is not an easy task to make them uniform, as

many records are quite old, concentrated during 1960-1980, and will require an extensive archeology effort. The contrast in the number of records in the surgedat dataset between 1960-1980, and 1980-2000 perhaps also deserves further dedicated research.

By comparing with our modelling results, we suspect that some of the values, particularly those located in the central delta, are probably wrong. For example, the 9m water level around Sunderban is extremely unlikely due to widespread mangrove areas there, which attenuate the surges. The tidal range is also lower there compared to both east and west corners. However, it is still interesting to note that the spatial density pattern of the storm tide record location has a high similarity with the landfall location distribution of our storm dataset.

Finally, due to the uncertainty with the datum, we refrained from reporting further results in our manuscript regarding this dataset.

**2) Population exposure**

I appreciate the efforts made by the authors to discuss the limitations of their approach. In addition to the limitations raised in Sect. 5.6, could the authors also consider / discuss the use of alternatives population dataset. For instance, the Global Human Settlement Layer - Population Grid r2019a has a spatial resolution of 9 arc sec, and the WorldPop Global High Resolution Population Denominators has a spatial resolution of 3 arc sec. See references below.

Florczyk, A. J., Corbane, C., Ehrlich, D., Freire, S., Kemper, T., Maffenini, L., Melchiorri, M., Pesaresi, M., Politis, 1715 P., Schiavina, M., and others: GHSL data package 2019, 29788, 290498, 2019

Lloyd, C. T., Chamberlain, H., Kerr, D., Yetman, G., Pistolesi, L., Stevens, F. R., Gaughan, A. E., Nieves, J. J., Hornby, G., MacManus, K., Sinha, P., Bondarenko, M., Sorichetta, A., and Tatem, A. J.: Global spatio-temporally harmonised datasets for producing high- resolution gridded population distribution datasets, Big Earth Data, 3, 108– 1780 139, https://doi.org/10.1080/20964471.2019.1625151, 2019.

Reply:

We thank the reviewer for suggesting these datasets. We have now adopted the GHSL dataset for our analyses, which has a resolution equivalent to the resolution of our model in the coastal regions (250m).

Adopting this new dataset reduced the population estimate under exposure to 5-year flood, as well as the increase of population count from 5-year to 50-year flood. But The overall 'percentage' of population under exposure to a 50-year flood remains the same as before, i.e., 10%.

Section 5.6 has been fully revised as following -

"...with a grey colour bar based on GHS Population dataset of 2015 (Schiavina et al., 2019). This dataset is based on GPWv4 dataset (Center For International Earth Science Information Network (CIESIN), Columbia University, 2016) illustrated in Figure 1, but disaggregated from the original administrative/census level data to grid cells based distribution of the built-up areas (Freire et al., 2016). The contours of the flooding extent at return period ranging from 5-year to 500-year are shown in colour.

As the population data is provided at a regular longitude-latitude grid, and the model grid is unstructured, it is first necessary…"

"... the same regular longitude-latitude grid as the population dataset (250m resolution). As the model grid …"

"The estimated population living in our model domain over the Bengal delta extent shown in Figure 9a amounts to 32 million. This count amounts to the fraction of Bangladesh population living at an elevation 5m or less. The estimated size of the exposed population at various return periods of inundation is shown in Figure 9b. The shaded region in Figure 9b is the 95% confidence interval of the exposed population estimate. These estimates correspond to the population exposed to the 95% confidence interval of the water level estimates. Our estimate shows that about 1 million people currently live within the 5-year return period flood level area. Even if the embankments were to work without failure during a cyclone, about 2.5 million more people [95% confidence interval: 2.3, 2.9] would get exposed to the flooding of 50-year return period. This additional count of the population represents about 8% of the total population living inside the study area. At a 100-year return level, the fraction of exposed people increases to about 16% of the total population inside the modelled domain.

In this assessment, we did not consider the probable (although not publicly documented) existence of city protection embankments at local scale, which may distort the patterns of Figure 7a locally. We did not consider either the potential degradation of the earthen dikes, and possible dike breaching during an intense cyclonic event. Knowledge of these factors will surely impact the anticipated exposure of the population to flooding suggested by our analysis. Instead, our focus was mainly on the physical mechanism of the flooding from storm surges. Despite these limitations, our exposed population map provides useful and spatially continuous information at relevant spatial scales to document the exposure to storm surge flooding and to better understand the environmental risks to the vast, densely populated Bengal delta continuum."

**3) Use of JTWC dataset**

3.1. As far as I understand the use of JTWC dataset for Fig. 4 is not a validation per se but the objective is to show the consistency of the ensemble results. However we note some discrepancies in Fig. 4(b) and (c) that deserve some additional comments or clarifications.

In particular, the frequency for April drastically differs between JTWC and the ensemble approach. Adding some errorbars to these histograms may here also help nuancing these differences.

Reply: We have revised the figure with errorbars (Standard Error) for JTWC dataset. Before going into further discussion, we recall here that the number of JTWC events is 42 only. The revised figure is shown in Figure C4-

[Figure]

Figure C4. a) Spatial distribution of the paths of the cyclones that make landfall along the coast of Bengal delta. Each square bin is 20 km wide. A small subset of cyclones trajectories is shown in the inset. (b) Distribution of maximum wind speed of the synthetic cyclones compared to the JTWC dataset, (c) Annual distribution of the occurrence of the synthetic cyclones compared to JTWC dataset. In (b) and (c), errorbar indicates the standard error associated to the small sample size in JTWC dataset.

The errorbars in the JTWC values are computed based on the assumption that the number of events occurring in a month is a Poisson process. The location parameter in Poisson distribution (lambda, here the number of events) for each month is computed by multiplying the probability density and number of JTWC events. Then from 10000 values generated from this distribution, the standard deviation is computed - which is the standard error reported here.

A second approach is also tested, using a bootstrap method, where a large set of samples (10000) of the same size as the JTWC dataset (42) is pulled from our ensemble. Then the monthly distribution is computed for the 10000 set, and the standard error is computed as the standard deviation of the 10000 instances for each month. Both the assumption of Poisson distribution and bootstrap method gives essentially similar results (Figure C).

[Figure]

Figure C5. Same as C4 but the uncertainty range is computed using bootstrap method.

We propose to provide a description of these computations in the Supplementary materials.

From both of these figures, what we see is that the small sample size of the JTWC dataset largely explains the variation.

The text was also updated accordingly to incorporate these changes -

"...The consistency of the cyclone ensemble is illustrated through a comparison with the observed JTWC statistics for maximum wind speed (Vmax) (Figure 4b) and seasonal distribution (Figure 4c). Errorbars in both cases indicate the standard error of the observation (i.e., short-length of dataset) given the probability distribution in the ensemble computed assuming a Poisson process. A similar standard error estimate was obtained using bootstrap method (See Supplementary materials).

The distribution of the simulated maximum wind speed (Vmax ) shows a good agreement with the observations from JTWC (Figure 4b). The standard errors for the JTWC dataset indicate that the ensemble distribution of Vmax is within the range of observational uncertainty.

Similarly, the seasonal distribution of the cyclone ensemble and that observed from JTWC both show a well matching pattern with a bimodal seasonality (Figure 4c) (Alam and Dominey-Howes, 2014). In the Bay of Bengal, low-pressure systems typically cannot intensify into a storm due to strong vertical wind shear present during monsoon (June-August). During the pre-monsoon (March-May) and post-monsoon (September-November), low vertical wind shear, and high sea surface temperature provide a suitable condition for low-pressure systems to intensify. For all months, except April, the ensemble cyclone density is typically within the range of observational uncertainty indicated by the errorbar (Figure 4c). However, for our storm surge hazard analysis, no measurable impact is expected from such intra-seasonal differences in cyclone distribution. The overall simulated bimodal temporal evolution of the synthetic cyclone indicates that the temporal statistics captured by our statistical-deterministic method correspond well with the seasonal climatic characteristics."

**3.2. Is JTWC dataset used to estimate the average annual frequency of 0.70314 (indicated in line 285)? If so, please specify.**

Reply: The model tuning is largely based on the JTWC dataset. The track generation method used in this study (Emanuel 2006) is adjusted by a calibration factor. This calibration constant is the rate at which a random cyclone is generated from a distribution of historical genesis points. In our case, the calibration factor was set to 1.2, and subsequently the average annual frequency of the cyclone was found to be 0.7.

We propose to revise L285 as following to incorporate this information -

"The cyclone generation model was adjusted using a calibration factor based on the observed cyclone genesis and displacement characteristics in the JTWC dataset. The calibration factor was set to 1.2, which indicates the rate of seeding for cyclone genesis in the probabilistic cyclone generation model. With an average annual frequency of 0.7 cyclones, the ensemble of 3600 cyclones considered here represents more than 5000 years of cyclonic activity over the northern Bay of Bengal under present climate conditions."

3.3. On page 19, in line 400, the authors state that 'This landfall pattern corresponds to previous observations that the landfalling cyclones in the Bangladesh coastline tend to move north-eastward'. Is this result also confirm by the analysis of the JTWC dataset?

Reply: This statement is supported by the JTWC dataset and discussed in further detail supplemented with numerical modelling in a recent article by Akter and Tsuboki (2021), for the storms recorded during the 1990-2019 period. The same conclusion is also drawn by Mondal et al. (2021).  We updated our manuscript to add these references as following -

"This landfall pattern corresponds to previous observations that the landfalling cyclones in the Bangladesh coastline tend to move north-eastward (Ali 1996, Akter and Tsuboki 2021, Mondal et al. 2021)."

**References**

Akter, N. and Tsuboki, K., 2021. Recurvature and movement processes of tropical cyclones over the Bay of Bengal. Quarterly Journal of the Royal Meteorological Society, 147(740), pp.3681-3702.

Mondal, M., Biswas, A., Haldar, S., Mandal, S., Bhattacharya, S. and Paul, S., 2021. Spatio-temporal Behaviours of Tropical Cyclones over the Bay of Bengal Basin in last Five Decades. Tropical Cyclone Research and Review. doi: 10.1016/j.tcrr.2021.11.004